# FSEO: A FEW-SHOT EVOLUTIONARY OPTIMIZATION FRAMEWORK FOR EXPENSIVE MULTI-OBJECTIVE OPTIMIZATION AND CONSTRAINED OPTIMIZATION

## ABSTRACT

Meta-learning has been demonstrated to be useful to improve the sampling efficiency of Bayesian optimization (BO) and surrogate-assisted evolutionary algorithms (SAEAs) when solving expensive optimization problems (EOPs). However, existing studies focuses on only single-objective optimization, leaving other expensive optimization scenarios unconsidered. We propose a generalized few-shot evolutionary optimization (FSEO) framework and focus on its performance on two common expensive optimization scenarios: multi-objective EOPs (EMOPs) and constrained EOPs (ECOPs). We develop a novel meta-learning modeling approach to train surrogates for our FSEO framework, an accuracy-based update strategy is designed to adapt surrogates during the optimization process. The surrogates in FSEO framework combines neural network with Gaussian Processes (GPs), their network parameters and some parameters of GPs represent useful experience and are meta-learned across related optimization tasks, the remaining GPs parameters are task-specific parameters that represent unique features of the target task. We demonstrate that our FSEO framework is able to improve sampling efficiency on both EMOP and ECOP. Empirical conclusions are made to guide the application of our FSEO framework.

## 1 INTRODUCTION

Expensive optimization problems (EOPs) aim to find as good as possible solutions within a budget of limited solution evaluations. Conventional Bayesian optimization (BO) and surrogate-assisted evolutionary algorithms (SAEAs) have been widely used to solve EOPs, but they train surrogate models from the scratch. To further improve the sampling efficiency and optimization performance, many efforts have been made to pre-train surrogates with the prior experience gain from related optimization tasks, resulting in experience-based optimization algorithms Bai et al. (2023); Liu et al. (2017); Tang et al. (2021); Tan et al. (2021).

This work considers solving EOPs on the context of few-shot problems Chen et al. (2019); Wang et al. (2020), where plenty of expensive related tasks are available and each of them can provide a small dataset for experience learning. Therefore, many experience-based optimization approaches such as multi-tasking optimization Wei et al. (2021); Bali et al. (2019); Xue et al. (2020), transfer optimization Tan et al. (2021); Jiang et al. (2020b;a) are not considered as they cannot learn experience from small related tasks (A discussion is available in Appendix A.1). In comparison, meta-learning Hospedales et al. (2021) has been proved to be powerful in solving few-shot problems, leading to a new subcategory of experience-based optimization, namely few-shot optimization (FSO) Wistuba & Grabocka (2021).

Existing studies on FSO are mainly few-shot Bayesian optimization (FSBO) where meta-learning approaches are combined with BO to solve EOPs with only one objective. These studies either employ meta-learning models from the literature directly or focus on the meta-learning of acquisition functions (AFs) that are customized for BO. In this paper, we develop a novel meta-learning architecture to enhance modeling performance and propose a generalized few-shot evolutionary optimization (FSEO) framework to address EOPs from the perspective of SAEAs. We demonstrate the generality and applicability of FSEO by considering two popular expensive optimization sce-

narios which have been limited studied but have a higher requirement on modeling performance than expensive single-objective optimization: multi-objective EOPs (EMOPs) and constrained EOPs (ECOPs). Major contributions are summarized as follows.

- A novel meta-learning method, namely Meta Deep Kernel Learning (MDKL), is developed to gain prior experience from related expensive tasks. Our model architecture and parameter designs make it possible to generate a regression-based surrogate on the prior experience and then continually adapt the surrogate to approximate the target task.
- We propose a FSEO framework to solve EOPs from the perspective of SAEAs. FSEO framework is applicable to regression-based SAEAs since FSEO embed our meta-learning models in these SAEAs as their surrogates. In addition, an update strategy is designed to adapt surrogates constantly during the optimization. Note that our FSEO framework is a general framework but we focus on its performance on EMOPs and ECOPs in this paper.
- Experiments are conducted on EMOPs and ECOPs to show our FSEO framework is effective. Our comprehensive ablation studies discover the influence of some factors on FSEO performance and provide empirical guidance to the application of FSEO framework.

## 2 RELATED WORK

Experience-based optimization can be divided into several subcategories according to the techniques of learning prior experience from related tasks. A detailed classification and discussion on these subcategories is available in Appendix A.1. This subsection focuses on related work on FSO.

FSO studies in the literature can be classified based on their model architectures. Most studies meta-learn parameters for Gaussian Processes (GPs) Williams & Rasmussen (2006), namely FSBO or Meta Bayesian Optimization (MBO) Shala et al. (2023); Wang et al. (2021); Pan et al. (2023); Tighineanu et al. (2023). In addition, Maraval et al. (2023) meta-learns with transformer neural processes and Wistuba & Grabocka (2021); Chen et al. (2023) meta-learn parameters for the architecture of deep kernel learning (DKL) Wilson et al. (2016). The MDKL model in our FSEO belongs to the last category as its model architecture is relevant to DKL.

Our work is different from existing studies in three points: Firstly, the novel architecture of meta-learning model for optimization purpose. Many studies Wistuba & Grabocka (2021) use existing meta-learning models Patacchiola et al. (2020) as their surrogates. During the optimization process, these surrogates make predictions with newly observed data, which is a kind of data adaptation rather than model parameter adaptation. The parameters in these models are trained and fixed before the optimization process begins, no further parameter adaptations are made during the optimization since these meta-learning models are originally designed for regression or classification tasks rather than optimization tasks. In comparison, we develop a meta-learning model, MDKL, for optimization purpose. MDKL has novel model architecture with explicit task-specific parameters, which allows continual model parameter adaptations and thus improves modeling performance during the optimization. Secondly, the generality and broad applicability of FSEO. Existing works are mainly customized for specific algorithms or optimization problems. For example, the meta-learning settings for AFs Watanabe et al. (2023) are not applicable to the SAEAs without AFs. However, our FSEO work on the meta-learning of surrogates and it is applicable to various SAEAs, so our work widens the scope of existing FSO research. A detailed discussion between BO and SAEA is presented in Appendix A.2. In addition, most existing FSO studies investigated only global optimization, leaving other optimization scenarios such as EMOP and ECOP still awaiting for investigation. In contrast, as our MDKL is designed for optimization and is capable of continually adaptation, we pay attention on EMOPs and ECOPs which require more effective models than global optimization. Lastly, in-depth ablation studies are lacking in the literature, making it unclear which factors affect the performance of FSO. Our extensive ablation studies fill this gap and we conclude some empirical rules to improve the performance of FSO.

## 3 BACKGROUND

This section gives preliminaries about meta-learning and DKL. The former is the method of experience learning, the latter is the underlying structure of experience representation.

## 3.1 META-LEARNING IN FEW-SHOT PROBLEMS

In the context of few-shot problems, we have plenty of related tasks, each task $T$ contributes a couple of small datasets $D = \{(S, Q)\}$, namely support dataset $S$ and query dataset $Q$, respectively. After learning from datasets of random related tasks, a support set $S_*$ from new unseen task $T_*$ is given and one is asked to estimate the labels or values of a query set $Q_*$. The problem is called 1-shot or 5-shot when only 1 data point or 5 data points are provided in $S_*$. A comprehensive definition of few-shot problems is available in Chen et al. (2019); Wang et al. (2020).

Meta-learning methods have been widely used to solve few-shot problems Wang et al. (2020). They learn domain-specific features that are shared among related tasks as experience, such experience is used to understand and interpret the data collected from new tasks encountered in the future.

## 3.2 DEEP KERNEL LEARNING (DKL)

DKL aims at constructing kernels that encapsulate the expressive power of deep architectures for GPs. To create expressive and scalable closed form covariance kernels, DKL combines the non-parametric flexibility of kernel methods and the structural properties of deep neural networks. In practice, a deep kernel $k(\mathbf{x}^i, \mathbf{x}^j|\boldsymbol{\gamma})$ transforms the inputs $\mathbf{x}$ of a base kernel $k(\mathbf{x}^i, \mathbf{x}^j|\boldsymbol{\theta})$ through a non-linear mapping given by a deep architecture $\phi(\mathbf{x}|\mathbf{w}, \mathbf{b})$:

$$k(\mathbf{x}^i, \mathbf{x}^j|\boldsymbol{\gamma}) = k(\phi(\mathbf{x}^i|\mathbf{w}, \mathbf{b}), \phi(\mathbf{x}^j|\mathbf{w}, \mathbf{b})|\boldsymbol{\theta}), \tag{1}$$

where $\boldsymbol{\theta}$ and $(\mathbf{w}, \mathbf{b})$ are parameter vectors of the base kernel and the deep architecture, respectively. $\boldsymbol{\gamma} = \{\boldsymbol{\theta}, \mathbf{w}, \mathbf{b}\}$ is the set of all parameters in this deep kernel. Note that in DKL, all parameters $\boldsymbol{\gamma}$ of a deep kernel $k(\mathbf{x}^i, \mathbf{x}^j|\boldsymbol{\gamma})$ are learned jointly by using the log marginal likelihood function of GPs as a loss function. Such a jointly learning strategy has been shown to make a DKL algorithm outperform a combination of a deep neural network and a GP model, where a trained GP model is applied to the output layer of a trained deep neural network Wilson et al. (2016).

## 3.3 META-LEARNING ON DKL

An important distinction between DKL algorithms and the applications of meta-learning to DKL is that DKL algorithms learn their deep kernels from single tasks instead of collections of related tasks. Such a difference alleviates two drawbacks of single task DKL Tossou et al. (2019): First, the scalability of deep kernels is no longer an issue as datasets in meta-learning are small. Second, the risk of overfitting is decreased since diverse data points are sampled across tasks.

# 4 FEW-SHOT EVOLUTIONARY OPTIMIZATION (FSEO) FRAMEWORK

In this paper, $T_*$ denotes the target optimization task, and plenty of small datasets $D_i$ sampled from related tasks $T_i$ are available for experience learning. A complete list of notations is available at the beginning of Appendix.

## 4.1 OVERALL WORKING MECHANISM

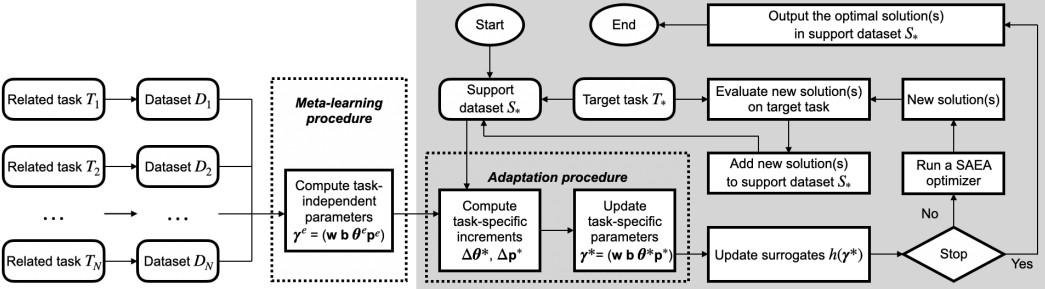

Figure 1: Diagram of our FSEO framework. Methods for handling multiple objectives or constraints are dependent on the module 'SAEA optimizer'.

As illustrated in Fig. 1, all modules covering the optimization of target task $T_*$ are included in a grey block. The modules beyond the grey block are associated with related tasks $T_i$ and experience learning, which distinguishes our FSEO framework from conventional SAEAs and BO. The MDKL surrogate modeling method consists of two procedures: meta-learning procedure and adaptation procedure. The former learns prior experience from $T_i$, the latter uses experience to adapt surrogates to fit $T_*$. The framework of FSEO is depicted in Alg. 1, it consists of the following major steps.

1. **Experience learning**: Before expensive optimization starts, a meta-learning procedure is conducted to train task-independent parameters $\gamma^e$ for MDKL surrogates (line 2). $N_m$ datasets $\{D_{m1}, \ldots, D_{mN_m}\}$ collected from $N$ related tasks $\{T_1, \ldots, T_N\}$ are used to train $\gamma^e$. $\gamma^e$ is the experience that represents the domain-specific features of related tasks.

2. **Initialize surrogates with experience**: Optimization starts when a target optimization task $T_*$ is given. An initial dataset $S_*$ is sampled (line 3) to adapt task-specific parameters $\gamma^*$ on the basis of experience $\gamma^e$. After that, MDKL surrogates are updated (line 4).

3. **Reproduction**: MDKL surrogates $h(\gamma^*)$ are combined with a SAEA optimizer $Opt$ to search for optimal solution(s) $\mathbf{x}^*$ on $h(\gamma^*)$ (line 7). This is implemented by replacing the original (regression-based) surrogates in a SAEA with $h(\gamma^*)$.

4. **Update archive and surrogates**: New optimal solution(s) $\mathbf{x}^*$ is evaluated on target task $T_*$ (line 8). The evaluated solutions will be added to dataset $S_*$ (line 9) which serves as an archive. Then, surrogate adaptation is triggered, surrogates $h(\gamma^*)$ are updated (line 10).

5. **Stop criterion**: Once the evaluation budget has run out, the evolutionary optimization will be terminated and the optimal solutions in dataset $S_*$ will be outputted. Otherwise, the algorithm goes back to step 3.

---

**Algorithm 1** FSEO Framework.

---

1: **Input:** $D_i$: Datasets collected from related tasks $T_i$, i=$\{1, \ldots, N\}$; $N_m$: Number of subsets $D_m$ for meta-learning; $|D_m|$: Size of subsets $D_m$. $|D_m| \leq |D_i|$ due to $D_m \subseteq D_i$; Batch size $B$; Surrogate learning rates $\alpha, \beta$; Target task $T_*$; A SAEA optimizer $Opt$; Fitness evaluation budget $FE_{max}$.
2: Experience $\gamma^e \leftarrow$ Meta-learning($D_i, N_m, |D_m|, B, \alpha$). /*Alg. 2.*/
3: $S_* \leftarrow$ Sampling $1d$ solutions from $T_*$.
4: $h(\gamma^*) \leftarrow$ Adaptation($\gamma^e, S_*, \beta$). /*Initialize surrogate.*/
5: Set evaluation counter $FE = |S_*|$.
6: **while** $FE < FE_{max}$ **do**
7:     Candidate solution(s) $\mathbf{x}^* \leftarrow$ Surrogate-assisted optimization ($Opt, h(\gamma^*)$).
8:     $f(\mathbf{x}^*) \leftarrow$ Evaluate $\mathbf{x}^*$ on $T_*$.
9:     $S_* \leftarrow S_* \cup \{(\mathbf{x}^*, f(\mathbf{x}^*))\}$.
10:     $h(\gamma^*) \leftarrow$ Update($\gamma^*, S_*, \beta$). /*Alg. 4.*/
11:     Update $FE$.
12: **end while**
13: **Output:** Optimal solutions in $S_*$.

---

## 4.2 Learning and Using Experience by MDKL

In MDKL, the domain-specific features of related tasks are used as experience, which are represented by the task-independent parameters $\gamma^e$ learned across related tasks. To make MDKL more capable of expressing complex domain-specific features, the base kernel $k(\mathbf{x}^i, \mathbf{x}^j \mid \boldsymbol{\theta})$ in GP is combined with a neural network $\phi(\mathbf{w}, \mathbf{b})$ to construct a deep kernel (see Eq.(1)). The modeling of a MDKL model consists of two procedures: meta-learning procedure and adaptation procedure. To make a clear illustration, we introduce frameworks of two procedures and then explain them in detail.

**Meta-learning procedure: Learning experience**
Our MDKL model uses the kernel in Jones et al. (1998) as its base kernel:

$$k(\mathbf{x}^i, \mathbf{x}^j | \boldsymbol{\theta}, \mathbf{p}) = \exp(-\sum_{k=1}^{d} \theta_k |x_k^i - x_k^j|^{p_k}). \tag{2}$$

Therefore, the deep kernel will be:

$$k(\mathbf{x}^i, \mathbf{x}^j | \boldsymbol{\gamma}) = \exp(-\sum_{k=1}^{d} \theta_k |\phi(x_k^i | \mathbf{w}, \mathbf{b}) - \phi(x_k^j | \mathbf{w}, \mathbf{b})|^{p_k}), \tag{3}$$

where $\boldsymbol{\gamma} = \{\mathbf{w}, \mathbf{b}, \boldsymbol{\theta}, \mathbf{p}\}$ is a set of deep kernel parameters. $\phi, \mathbf{w}$ and $\mathbf{b}$ are neural network and its parameters (see Eq.(1)). $\boldsymbol{\theta}, \mathbf{p}$ are parameters of base kernel, details about alternative base kernels are available in Williams & Rasmussen (2006).

The aim of meta-learning procedure is to learn experience $\boldsymbol{\gamma}^e$ from related tasks $\{T_1, \ldots, T_N\}$, including neural network parameters $\mathbf{w}, \mathbf{b}$, and task-independent base kernel parameters $\boldsymbol{\theta}^e, \mathbf{p}^e$. The pseudo-code of meta-learning procedure is given in Alg. 2.

---

**Algorithm 2** Meta-learning($D_i, N_m, |D_m|, B, \alpha$)

---

1: **Input:** $D_i$: Datasets collected from related tasks $T_i$, i=$\{1, \ldots, N\}$; $N_m$: Number of subsets $D_m$ for meta-learning; $|D_m|$: Size of subsets $D_m$. $|D_m| \leq |D_i|$ due to $D_m \subseteq D_i$; Batch size $B$; Learning rate for priors $\alpha$.
2: Randomly initialize $\mathbf{w}, \mathbf{b}, \boldsymbol{\theta}^e, \mathbf{p}^e$.
3: Set the number of update iterations U = $N_m/B$.
4: **for** $j = 1$ to $U$ **do**
5:    $\{D'_1, \ldots, D'_B\} \leftarrow$ Randomly select a batch of datasets from $\{D_1, \ldots, D_N\}$.
6:    **for all** $D'_i$ in the batch **do**
7:       $D_{mi} \leftarrow$ A subset of size $|D_m|$ from $D'_i$.
8:       Initialize task-specific increment $\Delta\boldsymbol{\theta}^i, \Delta\mathbf{p}^i$.
9:       Compute task-specific parameters: $\boldsymbol{\theta}^i = \boldsymbol{\theta}^e + \Delta\boldsymbol{\theta}^i, \mathbf{p}^i = \mathbf{p}^e + \Delta\mathbf{p}^i$.
10:      Obtain deep kernel $k(\mathbf{x}^i, \mathbf{x}^j | \boldsymbol{\gamma})$ based GP: $h(\boldsymbol{\gamma})$, where $\boldsymbol{\gamma} = \{\mathbf{w}, \mathbf{b}, \boldsymbol{\theta}^i, \mathbf{p}^i\}$ (Eq.(3)).
11:      Compute the loss function $L(D_{mi}, h(\boldsymbol{\gamma}))$ (Eq.(4)).
12:    **end for**
13:    Update $\mathbf{w}, \mathbf{b}, \boldsymbol{\theta}^e, \mathbf{p}^e$ via gradient descent: $\alpha \bigtriangledown L(D_{mi}, h(\boldsymbol{\gamma}))$ (Eq.(6)).
14: **end for**
15: **Output:** Task-independent parameters: $\boldsymbol{\gamma}^e = \{\mathbf{w}, \mathbf{b}, \boldsymbol{\theta}^e, \mathbf{p}^e\}$.

---

Ideally, the experience $\boldsymbol{\gamma}^e$ is learned from plenty of ($N_m$) small datasets $D_m$ collected from different related tasks. However, in practice, the number of available related tasks $N$ may be much smaller than $N_m$. Hence, the meta-learning is conducted gradually over $U$ update iterations (line 3). During each update iteration, a small batch of related tasks contribute $B$ small datasets $\{D_{m1}, \ldots, D_{mB}\}$ for meta-learning purpose (lines 5 and 7). Note that if $N < N_m$, a related task $T_i$ can be used multiple times in the meta-learning procedure.

For a given dataset $D_{mi}$, we denote $\boldsymbol{\theta}^i = \boldsymbol{\theta}^e + \Delta\boldsymbol{\theta}^i$ and $\mathbf{p}^i = \mathbf{p}^e + \Delta\mathbf{p}^i$ as the task-specific kernel parameters, where $\Delta\boldsymbol{\theta}^i, \Delta\mathbf{p}^i$ are the distance we need to move from the task-independent parameters to the task-specific parameters (line 9). The loss function $L$ of MDKL is the likelihood function defined as follows Jones et al. (1998):

$$\frac{1}{(2\pi)^{n/2}(\sigma^2)^{n/2}|\mathbf{R}|^{1/2}} exp[-\frac{(\mathbf{y} - \mathbf{1}\mu)^T \mathbf{R}^{-1}(\mathbf{y} - \mathbf{1}\mu)}{2\sigma^2}], \tag{4}$$

where $|\mathbf{R}|$ is the determinant of the correlation matrix $\mathbf{R}$, each element in the matrix is computed through Eq.(3). $\mathbf{y}$ is the fitness vector of $D_{mi}$. Mean $\mu$ and variance $\sigma^2$ of the prior distribution can be estimated by:

$$\hat{\mu} = \frac{\mathbf{1}^T \mathbf{R}^{-1} \mathbf{y}}{\mathbf{1}^T \mathbf{R}^{-1} \mathbf{1}}, \qquad \hat{\sigma} = \frac{1}{n}(\mathbf{y} - \mathbf{1}\hat{\mu})^T \mathbf{R}^{-1}(\mathbf{y} - \mathbf{1}\hat{\mu}). \tag{5}$$

Experience $\boldsymbol{\gamma}^e = \{\mathbf{w}, \mathbf{b}, \boldsymbol{\theta}^e, \mathbf{p}^e\}$ is updated by gradient descent (line 13), take $\boldsymbol{\theta}^e$ as an example:

$$\boldsymbol{\theta}^e \leftarrow \boldsymbol{\theta}^e - \frac{\alpha}{B}\sum_{i=1}^{B} \bigtriangledown_{\boldsymbol{\theta}^e} L(D_{mi}, h(\boldsymbol{\gamma})). \tag{6}$$

After $U$ iterations, $\boldsymbol{\gamma}^e$ has been trained sufficiently by $N_m$ small datasets $D_m$ and will be used in target task $T_*$ later.

**Adaptation procedure: Using experience**

The meta-learning of experience $\boldsymbol{\gamma}^e$ enables MDKL to handle a family of related tasks in general. To approximate a specific task $T_*$ well, surrogate $h(\boldsymbol{\gamma}^e)$ needs to adapt task-specific increments $\Delta\boldsymbol{\theta}^*$ and $\Delta\mathbf{p}^*$ in the way described in Alg. 3. A diagram of the deep kernel implemented in our MDKL model is illustrated in Fig. 2: From Fig. 2, it is clear that task-independent parameters $\boldsymbol{\gamma}^e = \{\mathbf{w}, \mathbf{b}, \boldsymbol{\theta}^e, \mathbf{p}^e\}$ are trained on meta data $D_i$. During the optimization process, MDKL adapts task-specific increments $\Delta\boldsymbol{\theta}^*, \Delta\mathbf{p}^*$ (Algorithm 8, line 3) and combines them with experience $\boldsymbol{\theta}^e$, resulting in task-specific parameters $\boldsymbol{\theta}^*, \mathbf{p}^*$. Hence, the deep kernel parameter $\boldsymbol{\gamma}^* = \{\mathbf{w}, \mathbf{b}, \boldsymbol{\theta}^*, \mathbf{p}^*\}$ is available. By invoking equation 5, the prior distribution of MDKL is estimated for the following surrogate prediction procedure.

---

**Algorithm 3** Adaptation($\boldsymbol{\gamma}^*, S_*, \beta$)

1: **Input:** Current surrogate parameters $\boldsymbol{\gamma}^*$; A dataset $S_*$ sampled from target task $T_*$ (Archive); Learning rate for adaptation $\beta$.
2: **if** $\boldsymbol{\gamma}^* == \boldsymbol{\gamma}^e$ **then**
3:     Initialize task-specific increments $\Delta\boldsymbol{\theta}^*, \Delta\mathbf{p}^*$.
4:     Compute task-specific parameters: $\boldsymbol{\theta}^* = \boldsymbol{\theta}^e + \Delta\boldsymbol{\theta}^*$, $\mathbf{p}^* = \mathbf{p}^e + \Delta\mathbf{p}^*$.
5:     Obtain deep kernel $k(\mathbf{x}^i, \mathbf{x}^j | \boldsymbol{\gamma}^*)$ based GP: $h(\boldsymbol{\gamma}^*)$, where $\boldsymbol{\gamma}^* = \{\mathbf{w}, \mathbf{b}, \boldsymbol{\theta}^*, \mathbf{p}^*\}$ (Eq.(3)).
6: **end if**
7: Compute the loss function $L(S_*, h(\boldsymbol{\gamma}^*))$ (Eq.(4)).
8: Update $\Delta\boldsymbol{\theta}^*, \Delta\mathbf{p}^*$ using gradient descent: $\beta\triangledown L(S_*, h(\boldsymbol{\gamma}^*))$.
9: **Output:** Adapted MDKL $h(\boldsymbol{\gamma}^*)$.

---

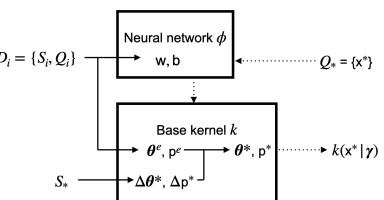

Figure 2: Diagram of our deep kernel implementation. The solid lines depict the training process, the dotted lines depict the inference process. $Q_*$ denotes query samples to be evaluated on our surrogates.

**Surrogate prediction.** Due to the nature of a GP, when predicting the fitness of a solution $\mathbf{x}^*$, a MDKL surrogate produces a predictive Gaussian distribution $\mathcal{N}(\hat{y}(\mathbf{x}^*), \hat{s}^2(\mathbf{x}^*))$, the predicted mean $\hat{y}(\mathbf{x}^*)$ and covariance $\hat{s}^2(\mathbf{x}^*)$ are specified as Jones et al. (1998):

$$\hat{y}(\mathbf{x}^*) = \mu + \mathbf{r}'\mathbf{R}^{-1}(\mathbf{y} - \mathbf{1}\mu), \tag{7}$$

$$\hat{s}^2(\mathbf{x}^*) = \sigma^2(1 - \mathbf{r}'\mathbf{R}^{-1}\mathbf{r}), \tag{8}$$

where $\mathbf{r}$ is a correlation vector consisting of covariances between $\mathbf{x}^*$ and $S_*$, other variables are explained in Eq.(4).

### 4.3 SURROGATE UPDATE STRATEGY

In this subsection, we describe the update strategy in our FSEO framework. To properly integrate experience and data from $T_*$, our update strategy is designed to determine whether a MDKL surrogate should be adapted in the current iteration or not, ensuring an optimal update frequency of surrogates.

As illustrated in Alg. 4, the surrogate update starts when a new optimal solution(s) has been evaluated on expensive functions and an updated archive $S_*$ is available. For a given surrogate $h(\boldsymbol{\gamma}^*)$, its mean squared error (MSE) on $S_*$ is selected as the update criterion: If the MSE after an adaptation $e_1$ (line 4) is larger than the MSE without an adaptation $e_0$ (line 2), then the surrogate will roll back to the status before the adaptation. This indicates the surrogate update has been refused and $h(\boldsymbol{\gamma}^*)$ will not be adapted in the current iteration. Otherwise, the adapted surrogate will be chosen (line 6). Note that no matter whether surrogate adaptations are accepted or refused, the resulting surrogates will be treated as updated surrogates, which are employed to assist the SAEA optimizer in the next iteration.

---

**Algorithm 4** Update($\boldsymbol{\gamma}^*, S_*, \beta$)

1: **Input:**
    Current surrogate parameters $\boldsymbol{\gamma}^*$;
    Updated archive $S_*$;
    Learning rate for further adaptations $\beta$.

2: $e_0 \leftarrow \text{MSE}(h(\boldsymbol{\gamma}^*), S_*)$.
3: $h(\boldsymbol{\gamma}') \leftarrow \text{Adaptation}(\boldsymbol{\gamma}^*, S_*, \beta)$.
    /*Temporary surrogate, Alg. 3.*/
4: $e_1 \leftarrow \text{MSE}(h(\boldsymbol{\gamma}'), S_*)$.
5: **if** $e_0 > e_1$ **then**
6:     update $\boldsymbol{\gamma}^* = \boldsymbol{\gamma}'$, obtain new $h(\boldsymbol{\gamma}^*)$.
7: **end if**
8: **Output:** Surrogate $h(\boldsymbol{\gamma}^*)$.

---

Table 1: Parameter setups for meta-learning methods.

| Module | Parameter | Value |
|---|---|---|
| Meta-learning | Number of meta-learning datasets $N_m$ | 20000 |
| | Number of update iterations $U$ | 2000 |
| | Batch size $B$ | 10 |
| Neural network | Number of hidden layers | 2 |
| | Number of units in each hidden layer | 40 |
| | Learning rates $\alpha, \beta$ | 0.001, 0.001 |
| | Activation function | ReLU |

Table 2: Parameter setups for DTLZ optimization.

| Parameter | MOEA/D-FS | Comparisons |
|---|---|---|
| Number of related tasks $N$ | 20000 ($N_m$ in Table 1) | - |
| Size of datasets from related tasks $|D_i|$ | 20 ($2d$) | - |
| Size of datasets for meta-learning $|D_m|$ | $|D_i|$ | - |
| Evaluations for initialization | 10 ($1d$) | 100 ($10d$) |
| Evaluations for further optimization | 50 | 50 |
| Total evaluations | 60 | 150 |

## 5 COMPUTATIONAL STUDIES

Our computational studies can be divided into three parts:

1. Appendix D evaluates our meta-learning model performance and analyzes model component contributions through a synthetic problem and a real-world engine modeling problem.
2. Sections 5.1 to 5.2 use EMOPs as examples to thoroughly investigate the performance of our FSEO framework in enhancing sampling efficiency. Empirical evidence is provided to support and guide the practical application of our FSEO framework.
3. Section 5.3 investigates the performance of FSEO framework in enhancing sampling efficiency for a real-world ECOP, demonstrating the broad applicability of FSEO framework.

The source code is available online[1] For all meta-learning methods used in our experiments, their basic setups are listed in Table 1. The neural network structure is suggested by Finn et al. (2017); Patacchiola et al. (2020), and the learning rates are the default values that have been widely used in many meta-learning methods Harrison et al. (2018); Patacchiola et al. (2020).

### 5.1 PERFORMANCE ON EMOPs

In the following subsections, we aim to demonstrate the effectiveness of our FSEO framework. The experiment in this subsection is designed to answer the question below: With the experience learned from related tasks, can our FSEO framework helps a SAEA to save $9d$ solutions without a loss of optimization performance?

The computational study is conducted on the DTLZ test problems Deb et al. (2005). All the DTLZ problems have $d = 10$ decision variables and 3 objectives, as the setups that have been widely used in Pan et al. (2018); Song et al. (2021). The details of generating DTLZ variants (related tasks) are provided in Appendix C. We test our FSEO framework using an instantiation on MOEA/D-EGO, resulting MOEA/D-FS. Details of the comparison algorithms are given in Appendix E.1.

#### 5.1.1 EXPERIMENTAL SETUPS

The parameter setups for this multi-objective optimization experiment are listed in Table 2. During the optimization process, an initial dataset $S_*$ is sampled using Latin-Hypercube Sampling (LHS) method McKay et al. (2000), then extra evaluations are conducted until the evaluation budget has run out. Note that we aim to use related tasks to save $9d$ evaluations without a loss of SAEA optimization performance. Hence, the total evaluation budgets for MOEA/D-FS and comparison algorithms are different.

Since the test problems have 3 objectives, we employ inverted generational distance plus (IGD+) Ishibuchi et al. (2015) as our performance indicator, where smaller IGD+ values indicate better optimization results. 5000 reference points are generated for computing IGD+ values, as suggested in Pan et al. (2018). More results in IGD Bosman & Thierens (2003) and HV Zitzler & Thiele (1998) metrics are reported in Appendix E.3.

#### 5.1.2 RESULTS AND ANALYSIS

The statistical test results are reported in Fig. 3 and Appendix E.2 (Table 5). It can be seen from Fig. 3 that, although 90 fewer evaluations are used in surrogate initialization, MOEA/D-FS can

---

[1]A link will be disclosed here once the paper is accepted.

still achieve competitive or even smaller IGD+ values than MOEA/D-EGO on all DTLZ problems except for DTLZ7. In addition, the IGD+ values obtained by MOEA/D-FS drop rapidly, especially during the first few evaluations, implying the experience learned from DTLZ variants are effective. Therefore, in most situations, our FSEO framework is able to assist MOEA/D-EGO in reaching competitive or even better optimization results, with the number of evaluations used for surrogate initialization reduced from $10d$ to only $1d$.

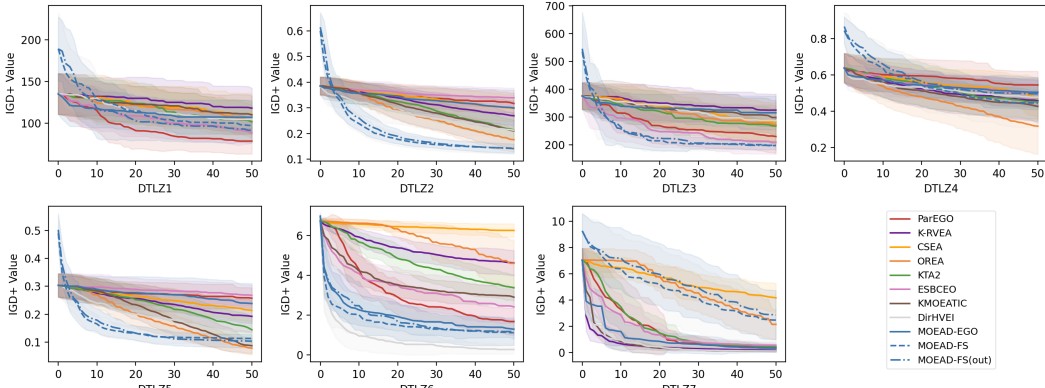

Figure 3: IGD+ curves averaged over 30 runs on the DTLZ problems. Solid lines are mean values, while shadows are error regions. MOEA/D-FSs and comparison algorithms initialize their surrogates with 10, 100 samples, respectively. X-axis denotes the extra 50 evaluations allowed in the further optimization. Note that 'FS(out)' indicates the target task is excluded from the range of related tasks during the meta-learning procedure) (see Appendix F).

MOEA/D-FS is less effective on DTLZ7 than on other DTLZ problems, which might be attributed to the discontinuity of the Pareto front on DTLZ7. Note that MOEA/D-FS learns experience from small datasets such as $D_m$ and $S_*$. The solutions in these small datasets are sampled at random, hence, the probability of having optimal solutions being sampled is small. However, it is difficult to learn the discontinuity of the Pareto front from the sampled non-optimal solutions. As a result, the knowledge of 'there are four discrete optimal regions' cannot be learned from such small datasets ($|D_m| = 20$) collected from related tasks. The performance analysis between MOEA/D-FS and other comparison algorithms are available in Appendix E.2.

### 5.1.3 More comparison experiments

We also compared the performance of our FSEO framework when only 10 evaluations are used for surrogate initialization for comparison algorithms. The results are reported in Table 8 in Appendix E.4. In addition, the performance of our FSEO framework in the context of extremely expensive optimization has been investigated in Appendix H (Table 11 and Fig. 8).

The question raised at the beginning of this subsection can be answered by the results discussed so far. Due to the integration of the experience learned from related tasks (DTLZ variants), although the evaluation cost of surrogates initialization has been reduced from $10d$ to $1d$, our FSEO framework is still capable of assisting regression-based SAEAs to achieve competitive or even better optimization results in most situations.

### 5.2 Ablation Studies and Performance on Real-World EMOPs

We conduct two ablation studies to investigate the influence of task similarity and that of the dataset size used in meta-learning, results and analysis are reported in Appendixes F and G, respectively.

We also test FSEO on a real-world EMOP: A network architecture search (NAS) problem. This problem optimizes the architecture of a Transformer in terms of two objectives: error and flops. The result is illustrated in Fig. 4, detailed experimental setups and analysis are available in Appendix I.

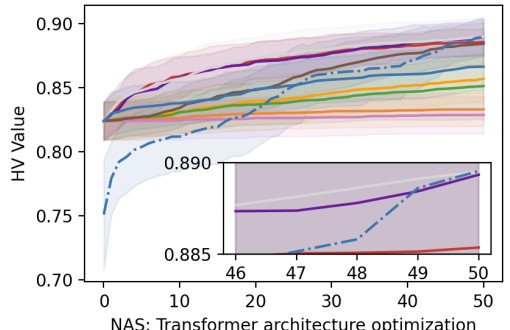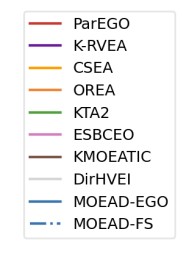

Figure 4: Results of 30 runs on a real-world NAS problem. Solid lines are mean values, while shadows are error regions. MOEA/D-FS and comparison algorithms initialize their surrogates with 18, 100 samples, respectively. X-axis denotes the extra 50 evaluations allowed in the further optimization. MOEA/D-FS reaches competitive results while 82 evaluations are saved.

### 5.3 PERFORMANCE ON REAL-WORLD ECOPs: ENGINE CALIBRATION

Our experiments on EMOPs have investigated the performance of our FSEO framework in depth. In this subsection, we evaluate our FSEO framework on an ECOP to demonstrate its generality and broad applicability, showing that sampling efficiency can be enhanced in both the objective space and constraint space. The ECOP is a real-world gasoline motor engine calibration problem with 6 adjustable engine parameters, namely throttle angle, waste gate orifice, ignition timing, valve timings, state of injection, and air-fuel-ratio. The calibration aims at minimizing BSFC while satisfying 4 constraints in terms of temperature, pressure, CA50, and load simultaneously Zhu et al. (2020).

#### 5.3.1 COMPARISON ALGORITHMS

Since the comparison algorithms in the DTLZ optimization experiments are not designed for handling constrained optimization, our comparison is conducted with 3 state-of-the-art constrained optimization algorithms used in industry: A variant of EGO designed to handle constrained optimization problems (cons_EGO) Zhu et al. (2020), a GA customized for this calibration problem (adaptiveGA) Zhu et al. (2020), and a bilevel constrained SAEA (SAB-DE) Yu et al. (2023). The settings of the comparison algorithms are the same as suggested in the literature. In this experiment, we apply our FSEO framework to cons_EGO and investigate its optimization performance. Specifically, we meta-learn MDKL surrogates for each objective and each constraint separately, and set the underlying optimization method as well as the constraint handling technique in cons_EGO as an optimizer in Fig. 1. The resulting algorithm is denoted as cons_FS in our experiments.

#### 5.3.2 EXPERIMENTAL SETUPS

The setup of related tasks $(N, D_i)$ is the same as described in Appendix D. In the meta-learning procedure, both the support set and the query set contain 6 data points, thus $|D_m| = 12$. The total evaluation budget for all algorithms is set to 60. For adaptiveGA, all evaluations are used in the optimization process as it is not a SAEA. For cons_EGO and SAB-DE, 40 samples are used to initialize the surrogates and 20 extra evaluations are used in the optimization process. For cons_FS, only 6 samples are used to initialize MDKL surrogates, and the remaining evaluations are used for further optimization.

#### 5.3.3 OPTIMIZATION RESULTS AND ANALYSIS

The left side and right side of Fig. 5 plot the normalized BSFC results and the number of feasible solutions found over the number of used evaluations, respectively. Solid lines are mean lines, while shadows are error regions. From the left side of Fig. 5, it can be observed that the minimal BSFC obtained by cons_FS decreases drastically in the first few evaluations, implying that the experience learned from related tasks is effective. In comparison, the minimal BSFC obtained by adaptiveGA

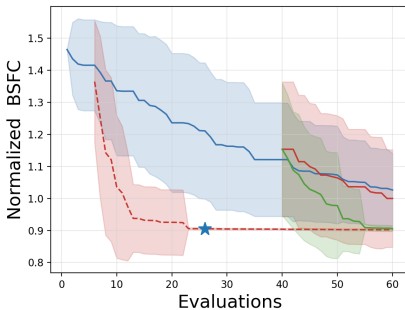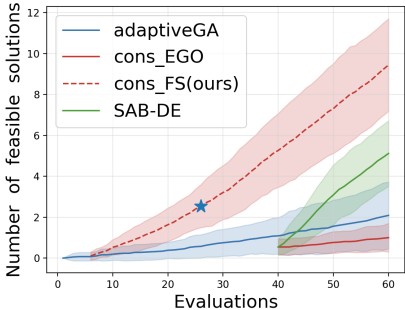

Figure 5: Results of 30 runs on the real-world engine calibration problem, all BSFC values are normalized. Solid lines are mean values, while shadows are error regions. Left figure shows how BSFC varies with the number of evaluations. The star markers highlight the results achieved when 20 evaluations are used in the optimization process. Right figure illustrates how the number of feasible solutions varies with the number of evaluations.

and cons_EGO drops in a relatively slow rate, even though cons_EGO has used 34 more samples to initialize its surrogates. The star marker denotes the point at which cons_FS has evaluated 20 samples after surrogate initialization. It is worth noting that when 20 samples have been evaluated in the optimization, cons_FS achieves a smaller BSFC value than cons_EGO. After the star marker, the decrease of BSFC becomes slow as cons_FS has reached the optimal region. Therefore, further improvement in the normalized BSFC value is not significant and thus hard to be observed. The advantages of our FSEO framework can also be observed in constraint handling. In the right side of Fig. 5, cons_FS finds more feasible solutions than the 3 comparison algorithms. These results indicate that our FSEO framework improves the performance of cons_EGO on both objective function and constraint functions. Meanwhile, only $1d$ evaluations are used to initialize surrogates.

**Discussion on runtime.** It should be noted that evaluating real engine performance on engine facilities is very costly in terms of both time and financial budget Yu et al. (2022). A single real engine performance evaluation can cost several hours Ma (2013), making the time cost of the meta-learning procedure negligible as it takes only a few minutes.

## 6 CONCLUSION AND FURTHER WORK

**Conclusion.** In this paper, we present a FSEO framework to address EMOPs and ECOPs from the perspective of SAEAs. A novel meta-learning approach MDKL is proposed to learn prior experience from related expensive tasks. Our MDKL model is designed for optimization and has explicit task-specific parameters, which allows continually update of task-specific parameters during the optimization process. Our empirical experiments show that the FSEO framework is able to improve the sampling efficiency and thus save expensive evaluations for existing regression-based SAEAs. Ablation studies reveal the influence between optimization performance and solutions similarity as well as the size of datasets for meta-learning.

**Limitation and future work.** The limitations of this work can be summarized as the following two points: First, we do not have a mathematical definition of related tasks. As a result, the boundary between related and unrelated tasks is not clear, making it difficult to conduct theoretical analysis on task similarity. Second, the proposed framework is currently for regression-based SAEAs only. A detailed discussion on this point is available in Appendix B. Future work could focus on quantifying task similarity by proposing a metric to measure the similarity between tasks. With an appropriate task similarity measure, systematic studies on few-shot optimization and experience-based optimization could be conducted. In addition, few-shot optimization framework for other SAEA categories can also be a future work.

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

# A  DISCUSSION ON RELEVANT CONCEPTS

## A.1  DIFFERENCE BETWEEN EXPERIENCE-BASED OPTIMIZATION

In the past decade, experience-based optimization has attracted much attention as it uses the experience gained from other optimization problems to improve the optimization efficiency of target problems, which mimics human capabilities of cognitive and knowledge generalization Gupta et al. (2017). The optimization problems that provide experience or knowledge are regarded as source tasks, while the target optimization problems are regarded as target tasks. To obtain useful experience, the tasks that are related to target tasks are chosen as source tasks since they usually share domain-specific features with target tasks. Diverse experience-based optimization methods have been proposed to use the experience gained from related tasks to tackle target tasks. They can be divided into two categories based on the direction of experience transformation.

In the first category, experience is transformed mutually. Every considered optimization problem is a target task and also is one of the source tasks of other optimization problems. In other words, the roles of source task and target task are compatible. One representative tributary is EMTO that aims to solve multiple optimization tasks concurrently Ding et al. (2017); Wei et al. (2021); Liaw & Ting (2019); Bali et al. (2019); Xue et al. (2020). In EMTO, experience is learned, updated, and spontaneously shared among target tasks through multi-task learning techniques. A variant of EMTO is multiforms optimization Gupta et al. (2017); Zhang et al. (2021); Guo et al. (2022). In multiforms optimization, multi-task learning methods are employed to learn experience from distinct formulations of a single target task.

In the second category, experience is transformed unidirectionally. The roles of source task and target task are not compatible, an optimization problem cannot serve as a source task and a target task simultaneously. One popular tributary is transfer optimization which employs transfer learning techniques to transform experience from source tasks to target tasks Tan et al. (2021); Jiang et al. (2020b;a); Volpp et al. (2020). In transfer learning, experience can be transformed from a single source task, multiple source tasks, or even source tasks from a different domain Zhuang et al. (2020). However, these transfer learning techniques pay more attention to experience transformation instead of experience learning. Despite diverse and complex situations of experience transformation have been studied Ruan et al. (2019; 2020), the difficult of learning experience from small (expensive) source tasks has not been well studied. Actually, a common scenario in transfer learning is that the source task(s) is/are large enough such that useful experience can be obtained easily through solving source task(s) Zhuang et al. (2020). In contrast to transfer optimization, recently, some experience-based optimization algorithms attempted to use meta-learning methods to learn experience from small source tasks, which are known as few-shot optimization (FSO)Wistuba & Grabocka (2021). Since meta-learning only works for related tasks in the same domain, the situations of experience transformation are less complex than that of transfer learning. As a result, meta-learning pays more attention to experience learning instead of experience transformation. Domain-specific features are extracted as experience and no related task needs to be solved.

Our work belongs to the FSO in the second category discussed above since our experience is transformed unidirectionally. More importantly, our experience is learned across many related expensive tasks, rather than gained through solving more or less source tasks. Therefore, our work is different from transfer optimization.

## A.2  DIFFERENCE BETWEEN BAYESIAN OPTIMIZATION AND SURROGATE-ASSISTED EVOLUTIONARY ALGORITHM

Bayesian Optimization (BO) and Surrogate-Assisted Evolutionary Algorithm (SAEA) are both model-based optimization methods for solving expensive optimization problems. The difference between BO and SAEA can be summarized as follows:

1. Surrogate models type. BO uses probabilistic models, such as GPs, as surrogates. In comparison, SAEAs are flexible and can use any type of approximation model, not limited to probabilistic models.

2. Selection criterion. BO designs an acquisition function (AF) as the selection criterion for candidate solutions, explicitly considering the uncertainty in the probabilistic models.

However, SAEAs do not necessarily account for model uncertainty. Instead, they focus on diversity and convergence as selection criteria, which can be implemented through separate functions.

3. Search algorithm. BO has no limitation on the search algorithm and can use either gradient-based or gradient-free optimization (such as EAs) to search candidate solutions. In contrast, SAEAs use only EAs as their underlying optimization algorithms.

As a result, there is some overlap between BO and SAEAs. A typical example is ParEGO, which employs GPs as its surrogates and designs an expected improvement (EI) function as its AF to consider uncertainty. Additionally, an EA is used as the underlying search algorithm.
Our FSEO framework focuses on meta-learning surrogates instead of AFs, making it compatible with various SAEAs that do not rely on model uncertainty or AFs as selection criteria. In comparison, existing studies mainly work on the meta-learning of AFs, which limits their generality and applicability to SAEAs.

## B  DISCUSSION ON FRAMEWORK COMPATIBILITY AND LIMITATION

Our FSEO framework is applicable to regression-based SAEAs as our MDKL surrogates can be embedded in these SAEAs directly. Classification-based SAEAs are not compatible with our FSEO framework. The classification surrogates in these SAEAs are employed to learn the relation between pairs of solutions, or the relation between solutions and a set of reference solutions. The class labels used for surrogate training can be fluctuating during the optimization and thus hard to be learned over related tasks. Similarly, in ordinal-regression-based SAEAs, the ordinal relation values to be learned are not as stable as the fitness of expensive functions. So ordinal-regression-based SAEAs are also not compatible with our FSEO framework. In this paper, we focus on FSO for regression-based SAEAs, while other SAEA categories are left to be discussed in future work.

## C  GENERATION OF DTLZ VARIANTS

Our DTLZ optimization experiments generate $m$-objective DTLZ variants in the following ways:
**DTLZ1**:

$$f_1 = (a_1 + g)0.5 \prod_{i=1}^{m-1} x_i, \tag{9}$$

$$f_{j=2:m-1} = (a_j + g)(0.5 \prod_{i=1}^{m-j} x_i)(1 - x_{m-j+1}), \tag{10}$$

$$f_m = (a_m + g)0.5(1 - x_1), \tag{11}$$

$$g = 100 \left( k + \sum_{i=1}^{k} \left( (z_i - 0.5)^2 - cos\left(20\pi(z_i - 0.5)\right) \right) \right), \tag{12}$$

where $\mathbf{z}$ is a vector consisting of the last $k = d - m + 1$ variables in $\mathbf{x}$. In other words, $\mathbf{z} = \{z_1, \ldots, z_k\} = \{x_m, \ldots, x_d\}$. The variants of DTLZ1 introduce only one variable $\mathbf{a} \in [0.1, 5.0]^m$ in Eq.(9), Eq.(10), and Eq.(11), where $\mathbf{a} = \mathbf{1}$ in the original DTLZ1. For out-of-range test, $\mathbf{a} \in [1.5, 5.0]^m$.

**DTLZ2**:

$$f_1 = (a_1 + g) \prod_{i=1}^{m-1} cos(\frac{x_i\pi}{b_1}), \tag{13}$$

$$f_{j=2:m-1} = (a_j + g) \left( \prod_{i=1}^{m-j} cos(\frac{x_i\pi}{b_j}) \right) sin(\frac{x_{m-j+1}\pi}{b_j}), \tag{14}$$

$$f_m = (a_m + g)sin(\frac{x_1\pi}{b_m}), \tag{15}$$

$$g = \sum_{i=1}^{k} (z_i - 0.5)^2. \tag{16}$$

The variants of DTLZ2 introduce two variables $\mathbf{a} \in [0.1, 5.0]^m$ and $\mathbf{b} \in [0.5, 2.0]^m$ in Eq.(13), Eq.(14), and Eq.(15), where $\mathbf{a} = \mathbf{1}$ and $\mathbf{b} = \mathbf{2}$ in the original DTLZ2. For out-of-range test, $\mathbf{a} \in [1.5, 5.0]^m, \mathbf{b} \in [0.5, 1.5]^m$.

**DTLZ3**: The variants of DTLZ3 are generated using the same way as described in DTLZ2, except the equation $g$ from Eq.(16) is replaced by the one from Eq.(12).

**DTLZ4**: The variants of DTLZ4 are generated using the same way as described in DTLZ2, except all $x_i$ are replaced by $x_i^{100}$.

**DTLZ5**: The variants of DTLZ5 are generated using the same way as described in DTLZ2, except all $x_2, \ldots, x_{m-1}$ are replaced by $\frac{1+2gx_i}{2(1+g)}$.

**DTLZ6**:

$$g = \sum_{i=1}^{k} z_i^{0.1}. \tag{17}$$

The variants of DTLZ6 are generated using the same way as described in DTLZ5, except the equation $g$ from Eq.(16) is replaced by the one from Eq.(17).

**DTLZ7**:

$$f_{j=1:m-1} = x_j + a_j, \tag{18}$$

$$f_m = (1 + g) \left( m - \sum_{i=1}^{m-1} \left( \frac{f_i}{1+g} (1 + sin(3\pi f_i)) \right) \right), \tag{19}$$

$$g = a_m + 9 \sum_{i=1}^{k} \frac{z_i}{k}. \tag{20}$$

The variants of DTLZ7 introduce one variable $\mathbf{a} \in [0.1, 5.0]^m$ in Eq.(18) and Eq.(20), where $a_{j=1:m-1} = 0$ and $a_m = 1$ in the original DTLZ7. For out-of-range test, $\mathbf{a} \in [1.5, 5.0]^m$.

## D  PERFORMANCE OF META-LEARNING MODEL AND COMPONENT CONTRIBUTION ANALYSIS

Evaluating the effectiveness of learning experiences aims to demonstrate that our MDKL model can learn experience from related tasks and outperforms other meta-learning models. For this reason, the experiment is designed to answer the following questions:

- Given a small dataset $S_*$ from target task $T_*$, can MDKL learn experience from related tasks and then generate a model that has the smallest MSE?

- If yes, which components of MDKL contribute to the effectiveness of learning experience? Meta-learning or/and deep kernel learning? If not, why not?

To answer the two questions above, we consider two experiments to evaluate the effectiveness of learning experience: amplitude prediction for unknown periodic sinusoid functions, and fuel consumption prediction for a gasoline motor engine. The former is a few-shot regression problem that motivates many meta-learning studies Finn et al. (2017); Harrison et al. (2018); Tossou et al. (2019); Patacchiola et al. (2020), while the latter is a real-world regression problem Zhu et al. (2020).

### D.1 EFFECTIVENESS OF LEARNING EXPERIENCE: SINUSOID FUNCTION REGRESSION

In the sinusoid regression experiment, we learn experience from a series of 1-dimensional sinusoid functions:

$$y = A sin(wx + b) + \epsilon, \tag{21}$$

where the amplitude $A$ and phase $w$ of sine waves are varied between functions. The target is to approximate an unknown sinusoid function with a small support dataset $S_*$ and the learned experience. Clearly, by integrating experience with $S_*$, we estimate parameters $(A, w, b)$ for an unknown sinusoid function. As a result, the output $y$ of the given sinusoid function can be predicted once a query data $x$ is inputted.

#### D.1.1 GENERATION OF SINUSOID FUNCTION VARIANTS

As suggested in Finn et al. (2017); Harrison et al. (2018), we set amplitude $A \in [0.1, 5.0]$, frequency $w \in [0.999, 1.0]$, phase $b \in [0, \pi]$, and Gaussian noise $\epsilon \sim (0, 0.1)$. Therefore, a sinusoid function can be generated by sampling three parameters $(A, w, b)$ from their ranges uniformly. In total, $N_m = N = 20000$ related sinusoid functions are generated at random.

#### D.1.2 EXPERIMENTAL SETUPS

All data points $x$ are sampled from the range $\in$ [-5.0, 5.0]. In the meta-learning procedure, both support set and query set contain 5 data points. Hence, a dataset $D_i$ is sampled from each (related) sinusoid function $T_i$, and $|D_i| = |D_m| = 10$. Six experiments are conducted where $|S_*| = \{2, 3, 5, 10, 20, 30\}$ data points are sampled from the target function. Considering Gaussian noise $\epsilon$ could be relatively large when amplitude $A$ is close to 0.1, normalized mean squared error (NMSE) is chosen as a performance indicator. NMSE is measured using a dataset that contains 100 data points sampled uniformly from the $x$ range.

#### D.1.3 COMPARISON METHODS

In this experiment, three families of modeling methods are compared with our MDKL model:

- **Meta-learning methods** that were proposed for regression tasks: MAML Finn et al. (2017), ALPaCA Harrison et al. (2018), and DKT Patacchiola et al. (2020). The configurations of MAML, ALPaCA, and DKT are the same as suggested in the original literature.

- **Non-meta-learning method** that is widely used for regression tasks: the GP model. We choose a GP as a baseline since it is effective and more relevant to MDKL than other non-meta-learning modeling methods. We set the range of base kernel parameters in the GP model as $\theta \in [10^{-5}, 10]$ and $p \in [1, 2]$.

- **MDKL related methods** that are designed to investigate which components of MDKL contribute to the modeling performance: GP_Adam, DKL, and MDKL_NN. GP_Adam is a GP model fitted by Adam optimizer. The combination of GP_Adam and a neural network results in a kind of DKL algorithm. MDKL_NN is a meta-learning version of DKL, but it learns only neural network parameters through meta-learning and has no task-independent base kernel parameters.

#### D.1.4 RESULTS AND ANALYSIS

Table 3 reports the statistical test results of the NMSE values achieved by comparison algorithms in sinusoid function regression experiments. Each row lists the results obtained when the same number of fitness evaluations $|S_*|$ are used to train models. The results of Wilcoxon rank sum test between MDKL and other compared algorithms are listed in the last row. It can be observed that both MDKL and DKT have achieved the smallest NMSE values on all tests in the comparison with other meta-learning and non-meta-learning modeling methods.

Contributions of MDKL components are analyzed through statistical tests between MDKL related methods. The statistical test results between DKL and GP_Adam are 5/1/0, showing that DKL is preferable to GP_Adam when only a few data points are available for modeling. Hence, using a neural network to build a deep kernel for GP is able to enhance the performance of modeling. When

Table 3: Mean NMSE and standard deviation (in parentheses) of 30 runs on the amplitude regression of sinusoid function. GP Stein (1999) is a widely used surrogate in SAEAs, MAML Finn et al. (2017), ALPaCA Harrison et al. (2018), and DKT Patacchiola et al. (2020) are meta-learning methods. GP_Adam is a GP model fitted by Adam optimizer. DKL is a deep kernel learning algorithm that adds a neural network to GP_Adam. MDKL_NN applies meta-learning to DKL, but no task-independent base kernel parameters are shared between related tasks. Support data points are used to train non-meta surrogates or adapt meta-learning surrogates. '+', '≈', and '−' denote MDKL is statistically significantly superior to, almost equivalent to, and inferior to the compared modelling methods in the Wilcoxon rank sum test (significance level is 0.05), respectively. The last row counts the total win/tie/loss results. It shows that MDKL and DKT have lower NMSE than other models. The effectiveness of meta-learning on both the neural network and the base kernel has been demonstrated on this example.

| Support data points $|S_*|$ | GP Stein (1999) | GP_Adam | DKL | MDKL_NN | MDKL (ours) | DKT Patacchiola et al. (2020) | MAML Finn et al. (2017) | ALPaCA Harrison et al. (2018) |
|---|---|---|---|---|---|---|---|---|
| 2 | 1.63e-1(9.18e-2)≈ | 1.93e-1(9.72e-2)+ | 1.63e-1(9.05e-2)≈ | 1.57e-1(9.26e-2)≈ | 1.56e-1(9.49e-2) | 1.56e-1(9.49e-2)≈ | 2.09e-1(3.63e-1)≈ | 1.07e+0(2.57e+0)≈ |
| 3 | 1.27e-1(6.04e-2)≈ | 1.62e-1(6.53e-2)+ | 1.21e-1(5.96e-2)≈ | 1.16e-1(5.95e-2)≈ | 1.10e-1(6.20e-2) | 1.10e-1(6.20e-2)≈ | 2.09e-1(3.60e-1)≈ | 4.36e-1(8.57e-1)≈ |
| 5 | 6.76e-2(4.62e-2)≈ | 1.09e-1(5.61e-2)+ | 7.52e-2(4.40e-2)+ | 6.38e-2(3.91e-2)+ | 4.79e-2(3.73e-2) | 4.79e-2(3.70e-2)≈ | 2.08e-1(3.59e-1)+ | 4.31e-1(8.04e-1)+ |
| 10 | 1.70e-2(1.87e-2)≈ | 6.13e-2(4.58e-2)+ | 2.87e-2(1.89e-2)+ | 1.89e-2(1.61e-2)+ | 1.07e-2(1.16e-2) | 1.09e-2(1.17e-2)≈ | 2.08e-1(3.58e-1)+ | 6.59e-1(2.14e+0)+ |
| 20 | 5.42e-3(7.64e-3)+ | 3.92e-2(4.29e-2)+ | 9.64e-3(1.02e-2)+ | 5.24e-3(6.57e-3)+ | 2.57e-3(4.53e-3) | 2.63e-3(4.61e-3)≈ | 2.08e-1(3.58e-1)+ | 1.13e-1(3.39e-1)+ |
| 30 | 3.97e-3(7.40e-3)+ | 3.32e-2(4.18e-2)+ | 4.81e-3(6.68e-3)+ | 3.20e-3(5.85e-3)+ | 1.68e-3(3.61e-3) | 1.60e-3(3.39e-3)≈ | 2.08e-1(3.58e-1)+ | 7.59e-2(2.01e-1)+ |
| $+/\approx/-$ | 2/4/0 | 6/0/0 | 4/2/0 | 3/3/0 | -/-/- | 0/6/0 | 4/2/0 | 3/3/0 |

meta-learning technique is applied to DKL, the statistical test results between MDKL_NN and DKL are 3/3/0. The meta-learning of neural network parameters is necessary since it contributes to the performance of MDKL. Further statistical test between MDKL and MDKL_NN gives results of 3/3/0, indicating that the meta-learning of base kernel parameters is effective on this regression problem.

## D.2 Effectiveness of Learning Experience: Engine Performance Regression

In this subsection, we focus on a Brake Special Fuel Consumption (BSFC) regression task for a gasoline motor engine Zhu et al. (2020), where BSFC is evaluated on a gasoline engine simulation (denoted by $T_*$).

### D.2.1 Experimental setups

The related tasks $T_i$ used in our experiment are $N = 100$ gasoline engine models. These engine models have different behaviors when compared with $T_*$, but they share the basic features of gasoline engines. All related tasks and the target task have the same six decision variables. Each related task $T_i$ provides only 60 solutions, forming a dataset $D_i$. The size of datasets used for meta-learning $|D_m|$ is set to 40. Six tests are conducted where $|S_*| = \{2, 3, 5, 10, 20, 40\}$ data points are sampled from the real engine simulation $T_*$. MSE is chosen as an indicator of modeling accuracy, which is measured using a dataset consisting of 12500 data points that are sampled uniformly from the engine decision space. The comparison algorithms are the same as described in Appendix D.1.

### D.2.2 Results and analysis

The statistical test results of the MSE values achieved by comparison algorithms in BSFC regression experiments are summarized in Table 4. Each row lists the results obtained when the same number of fitness evaluations $|S_*|$ are used to train models. The results of Wilcoxon rank sum test between MDKL and other compared algorithms are listed in the last row. It can be observed that MDKL and MDKL_NN outperform other comparison modeling methods since they have achieved the smallest MSE on all tests.

Additional Wilcoxon rank sum tests have been conducted between MDKL related algorithms to answer our second question (results are not reported in Table 4). The statistical test results between DKL and GP_Adam are 1/5/0, indicating that the neural network in DKL makes some contributions to the performance of MDKL. The statistical test results between MDKL_NN and DKL are 6/0/0, demonstrating that the meta-learning of neural network parameters constructs a useful deep kernel and contributes to the improvement of modeling accuracy. However, there is no significant difference between the performance of MDKL and that of MDKL_NN, the meta-learning on base kernel parameters does not play a critical role on this engine problem. In comparison, the meta-learning on

Table 4: Mean MSE and standard deviation (in parentheses) of 30 runs on the regression of engine fuel consumption. Support data points are used to train non-meta surrogates or adapt meta-learning surrogates. All results are normalized since the actual engine data is unable to be disclosed. The symbols '+', '≈', '−' denote the win/tie/loss result of Wilcoxon rank sum test (significance level is 0.05) between MDKL and comparison modeling methods, respectively. The last row counts the total win/tie/loss results.

| Support data points $|S_*|$ | GP Stein (1999) | GP_Adam | DKL | MDKL_NN | MDKL (ours) | DKT Patacchiola et al. (2020) | MAML Finn et al. (2017) | ALPaCA Harrison et al. (2018) |
|---|---|---|---|---|---|---|---|---|
| 2 | 2.23e+1(3.20e+0)+ | 2.37e+1(6.30e+0)+ | 2.30e+1(5.87e+0)+ | 1.73e+1(6.33e+0)≈ | 1.72e+1(6.34e+0) | 1.81e+1(5.68e+0)≈ | 1.87e+1(6.37e+0)≈ | 1.91e+1(1.02e+1)≈ |
| 3 | 2.14e+1(3.74e+0)+ | 2.41e+1(1.38e+1)+ | 2.20e+1(3.74e+0)+ | 1.45e+1(7.13e+0)≈ | 1.45e+0(7.01e+0) | 1.55e+1(6.66e+0)≈ | 1.80e+1(4.69e+0)≈ | 2.13e+1(1.97e+1)≈ |
| 5 | 2.13e+1(3.27e+0)+ | 2.46e+1(1.00e+1)+ | 2.07e+1(3.95e+0)+ | 1.12e+1(6.65e+0)≈ | 1.10e+1(6.58e+0) | 1.21e+1(6.49e+0)≈ | 1.84e+1(6.05e+0)+ | 1.99e+1(2.29e+1)+ |
| 10 | 1.84e+1(1.89e+0)+ | 2.06e+1(1.19e+1)+ | 2.10e+1(5.79e+0)+ | 7.19e+0(4.82e+0)+ | 7.08e+0(4.77e+0) | 7.99e+0(4.87e+0)≈ | 1.70e+1(5.54e+0)+ | 1.38e+1(8.12e+0)+ |
| 20 | 1.56e+1(2.00e+0)+ | 2.38e+1(1.05e+1)+ | 1.76e+1(2.42e+0)+ | 5.03e+0(1.82e+0)≈ | 4.86e+0(1.71e+0) | 5.74e+0(1.91e+0)+ | 1.50e+1(2.59e+0)+ | 1.01e+1(5.52e+0)+ |
| 40 | 1.28e+1(2.03e+0)+ | 1.48e+1(7.35e+0)+ | 1.67e+1(3.73e+0)+ | 4.13e+0(7.90e-1)≈ | 4.00e+0(8.59e-1) | 4.92e+0(1.09e+0)+ | 1.45e+1(1.85e+0)+ | 8.01e+0(3.35e+0)+ |
| +/≈/− | 6/0/0 | 6/0/0 | 6/0/0 | 0/6/0 | -/-/- | 2/4/0 | 4/2/0 | 4/2/0 |

base kernel parameters is effective in sinusoid function regression experiments (see Appendix D.1). In addition, the statistical test results between MDKL_NN and MAML are 4/2/0. Considering that MAML is a neural network regressor learned through meta-learning, we can conclude that GP is an essential component of our MDKL. In summary, all components in MDKL are necessary, they all contribute to the effectiveness of learning experience.

The comparison experiments on sinusoid functions and the gasoline motor engine have demonstrated the effectiveness of our MDKL modeling method in the learning of experience. Given a small dataset of the target task, the model learned through MDKL method has the smallest MSE among all comparison models. Additionally, the investigation between MDKL and its variants shows that all components in MDKL have made their contributions to the effectiveness of learning experience. However, similar to other meta-learning studies Finn et al. (2017); Harrison et al. (2018), we have not defined the similarity between tasks. In other words, the boundary between related tasks and unrelated tasks has not been defined. This should be a topic of further study on meta-learning. Moreover, the relationship between task similarity and modeling performance has not been investigated. Instead, we study the relationship between task similarity and SAEA optimization performance in Section F, since our main focus is the surrogate-assisted evolutionary optimization.

# E  ADDITIONAL DETAILS ON EXPENSIVE MULTI-OBJECTIVE OPTIMIZATION

## E.1  COMPARISON ALGORITHMS

As explained in Section B, our FSEO framework is compatible with regression-based SAEAs. Hence, we select MOEA/D-EGO Zhang et al. (2010) as an example and replace its GP surrogates by our MDKL surrogates. The resulting algorithm is denoted as MOEA/D-FS. Note that it is not necessary to specially select a newly proposed regression-based SAEA as our example, our main objective is to save evaluations with experience and observe if there is any damage to the optimization performance caused by the saving of evaluations. Therefore, it does not make any difference which regression-based SAEA or BO we choose as our example. Additionally, to demonstrate the improvement of optimization performance caused by using experience on DTLZ problems is significant, several state-of-the-art SAEAs and MOBO are also compared as baselines, including ParEGO Knowles (2006), K-RVEA Chugh et al. (2016), CSEA Pan et al. (2018), OREA Yu et al. (2019), KTA2 Song et al. (2021), ESBCEO Bian et al. (2023), KMOEA-TIC Qin et al. (2023) and DirHVEI Zhao & Zhang (2023). Among these algorithms, ParEGO, K-RVEA, KTA2, KMOEA-TIC use regression-based surrogates, CSEA uses a classification-based surrogate, OREA employs an ordinal-regression-based surrogate, ESBCEO and DirHVEI are recently proposed MOBO.

We implemented the FSEO framework, MOEA/D-EGO, ParEGO, and OREA, while the code of K-RVEA, CSEA, KTA2, ESBCEO, and DirHVEI is available on PlatEMO Tian et al. (2017), an open source MATLAB platform for evolutionary multi-objective optimization. The code of KMOEA-TIC Qin et al. (2023) is obtained from its authors. To make a fair comparison, all comparison algorithms share the same initial dataset $S_*$ in an independent run. We also set $\theta \in [10^{-5}, 100]^d$ and $\mathbf{p} = \mathbf{2}$ for all GP surrogates as suggested in Song et al. (2021), these GP surrogates are implemented through DACE Sacks et al. (1989). Other configurations are the same as suggested in their original literature.

Table 5: Mean IGD+ values and standard deviation (in parentheses) obtained from 30 runs on the DTLZ problems. MOEA/D-FS and the comparison algorithms initialize their surrogates with 10, 100 samples, respectively. Extra 50 evaluations are allowed in the further optimization.

| Problem | MOEA/D-EGO | MOEA/D-FS (ours) | ParEGO | K-RVEA | KTA2 | CSEA | OREA | ESBCEO | KMOEA-TIC | DirHVEI |
|---|---|---|---|---|---|---|---|---|---|---|
| DTLZ1 | 1.07e+2(2.05e+1)+ | 9.70e+1(1.87e+1) | 7.82e+1(1.54e+1)− | 1.18e+2(2.45e+1)+ | 1.01e+2(2.38e+1)≈ | 1.10e+2(2.50e+1)+ | 1.02e+2(1.97e+1)≈ | 8.81e+1(1.18e+1)≈ | 1.10e+2(2.29e+1)+ | 1.04e+2(2.47e+1)≈ |
| DTLZ2 | 2.99e-1(7.01e-2)+ | 1.43e-1(2.29e-2) | 3.17e-1(4.12e-2)+ | 2.69e-1(5.97e-2)+ | 2.14e-1(3.84e-2)+ | 2.98e-1(5.25e-2)+ | 1.76e-1(4.69e-2)+ | 3.39e-1(3.78e-2)+ | 2.10e-1(7.10e-2)+ | 2.13e-1(4.58e-2)+ |
| DTLZ3 | 3.15e+2(6.04e+1)+ | 1.97e+2(1.64e+1) | 2.30e+2(5.99e+1)≈ | 3.24e+2(5.90e+1)+ | 2.67e+2(6.70e+1)+ | 2.82e+2(6.97e+1)+ | 2.72e+2(6.88e+1)+ | 2.09e+2(4.23e+1)+ | 2.98e+2(6.14e+1)+ | 2.86e+2(5.98e+1)+ |
| DTLZ4 | 5.04e-1(8.25e-2)≈ | 4.44e-1(1.35e-1) | 5.44e-1(7.58e-2)+ | 4.57e-1(1.14e-1)≈ | 4.51e-1(9.54e-2)≈ | 4.75e-1(1.09e-1)≈ | 3.18e-1(1.54e-1)− | 4.99e-1(7.37e-2)≈ | 4.26e-1(9.19e-2)≈ | 4.77e-1(1.11e-1)≈ |
| DTLZ5 | 2.39e-1(7.17e-2)+ | 1.13e-1(2.24e-2) | 2.58e-1(3.68e-2)+ | 1.92e-1(5.97e-2)+ | 1.44e-1(4.60e-2)+ | 2.14e-1(4.05e-2)+ | 7.84e-2(2.42e-2)− | 2.68e-1(3.62e-2)+ | 8.73e-2(2.77e-2)− | 1.36e-1(3.72e-2)+ |
| DTLZ6 | 1.29e+0(4.74e-1)≈ | 1.11e+0(5.71e-1) | 1.67e+0(6.77e-1)+ | 4.62e+0(6.42e-1)+ | 3.37e+0(6.71e-1)+ | 6.26e+0(3.40e-1)+ | 4.60e+0(1.19e+0)+ | 2.41e+0(7.97e-1)+ | 2.90e+0(5.34e-1)+ | 2.65e-1(2.16e-1)− |
| DTLZ7 | 3.31e-1(3.11e-1)− | 2.47e+0(1.89e+0) | 3.66e-1(1.31e-1)− | 1.74e-1(3.57e-2)− | 4.34e-1(2.20e-1)− | 4.17e+0(1.13e+0)+ | 2.14e+0(1.15e+0)≈ | 5.47e-1(2.46e-1)− | 9.44e-2(1.23e-2)− | 1.18e-1(2.85e-2)− |
| +/≈/− | 4/2/1 | -/-/- | 4/1/2 | 5/1/1 | 4/2/1 | 6/1/0 | 3/2/2 | 3/3/1 | 4/1/2 | 3/2/2 |

## E.2 RESULT TABLE AND ANALYSIS OF EXPENSIVE MULTI-OBJECTIVE OPTIMIZATION

The experience learned from related tasks makes MOEA/D-EGO more competitive when compared to other SAEAs and MOBO algorithms. The use of MDKL surrogates results in significantly smaller IGD+ values on DTLZ1, DTLZ2, DTLZ3, and DTLZ5 than before. As a result, MOEA/D-FS achieves the smallest IGD+ values on DTLZ2 and DTLZ3, and its optimization results on DTLZ1 and DTLZ5 are much closer to the best optimization results (e.g. results obtained by ParEGO and OREA) than MOEA/D-EGO. Although MOEA/D-FS does not achieve the smallest IGD+ values on all DTLZ problems, it should be noted that MOEA/D-FS is still the best algorithm among comparison SAEAs due to its overall performance. Table 5 shows that no comparison SAEA outperforms MOEA/D-FS on three DTLZ problems, but MOEA/D-FS outperforms all comparison SAEAs on at least three DTLZ problems. Furthermore, the IGD+ values of MOEA/D-FS are achieved with an evaluation budget of 60, while the IGD+ values of other SAEAs and MOBO algorithms are reached with a cost of 150 evaluations (see Table 2).

## E.3 RESULT TABLES AND FIGURES IN IGD AND HV METRICS

The performance of our method and the comparison algorithms are also evaluated on inverted generational distance (IGD) Bosman & Thierens (2003) and Hypervolume (HV) Zitzler & Thiele (1998) metrics.

Results in IGD values are reported in Table 6 and Fig. 6. A smaller IGD value indicates a better optimization result.

Table 6: Mean IGD values and standard deviation (in parentheses) obtained from 30 runs on 7 DTLZ problems. MOEA/D-FS and comparison algorithms initialize their surrogates with 10, 100 samples, respectively. Extra 50 evaluations are allowed in the further optimization. '+', '≈', and '−' denote MOEA/D-FS is statistically significantly superior to, almost equivalent to, and inferior to the compared algorithms in the Wilcoxon rank sum test (significance level is 0.05), respectively. The last row counts the total win/tie/loss results.

| Problems | MOEAD-EGO | MOEAD-FS | ParEGO | K-RVEA | KTA2 | CSEA | OREA | ESBCEO | KMOEATIC | DirHVEI |
|---|---|---|---|---|---|---|---|---|---|---|
| DTLZ1 | 1.07e+2(2.05e+1)+ | 9.70e+1(1.87e+1) | 7.82e+1(1.54e+1)− | 1.18e+2(2.41e+1)+ | 1.01e+2(2.34e+1)≈ | 1.10e+2(2.46e+1)+ | 1.02e+2(1.97e+1)≈ | 8.81e+1(1.18e+1)≈ | 1.10e+2(2.29e+1)+ | 1.04e+2(2.47e+1)≈ |
| DTLZ2 | 3.30e-1(7.23e-2)+ | 1.72e-1(2.41e-2) | 3.59e-1(2.82e-2)+ | 3.08e-1(4.93e-2)+ | 2.45e-1(3.57e-2)+ | 3.36e-1(3.96e-2)+ | 2.14e-1(4.10e-2)+ | 3.64e-1(3.29e-2)+ | 2.86e-1(6.31e-2)+ | 3.40e-1(3.62e-2)+ |
| DTLZ3 | 3.15e+2(6.04e+1)+ | 1.97e+2(1.64e+1) | 2.30e+2(5.99e+1)≈ | 3.24e+2(5.80e+1)+ | 2.67e+2(6.58e+1)+ | 2.82e+2(6.85e+1)+ | 2.72e+2(6.88e+1)+ | 2.09e+2(4.23e+1)+ | 2.98e+2(6.14e+1)+ | 2.86e+2(5.98e+1)+ |
| DTLZ4 | 7.51e-1(1.50e-1)≈ | 7.96e-1(2.25e-1) | 7.65e-1(1.14e-1)≈ | 5.94e-1(1.28e-1)− | 6.30e-1(1.51e-1)− | 7.00e-1(1.48e-1)− | 5.64e-1(2.01e-1)− | 6.70e-1(8.05e-2)− | 5.23e-1(8.60e-2)− | 6.87e-1(9.73e-2)− |
| DTLZ5 | 2.47e-1(7.21e-2)+ | 1.17e-1(2.08e-2) | 2.83e-1(3.13e-2)+ | 2.13e-1(5.55e-2)+ | 1.61e-1(4.60e-2)+ | 2.33e-1(3.65e-2)+ | 8.64e-2(2.48e-2)− | 2.83e-1(3.00e-2)+ | 1.18e-1(3.17e-2)≈ | 2.00e-1(5.26e-2)+ |
| DTLZ6 | 1.36e+0(4.10e-1)≈ | 1.18e+0(5.35e-1) | 1.78e+0(6.29e-1)+ | 4.63e+0(6.26e-1)+ | 3.37e+0(6.50e-1)+ | 6.26e+0(3.28e-1)+ | 4.61e+0(1.18e+0)+ | 2.45e+0(7.92e-1)+ | 2.92e+0(5.35e-1)+ | 5.29e-1(2.25e-1)− |
| DTLZ7 | 4.22e-1(3.16e-1)− | 2.56e+0(1.86e+0) | 5.34e-1(1.25e-1)− | 2.55e-1(4.36e-2)− | 5.54e-1(2.38e-1)− | 4.20e+0(1.11e+0)+ | 2.21e+0(1.11e+0)≈ | 6.21e-1(2.43e-1)− | 1.85e-1(1.81e-2)− | 2.22e-1(3.76e-2)− |
| +/≈/− | 4/2/1 | -/-/- | 3/2/2 | 5/0/2 | 4/1/2 | 6/0/1 | 3/2/2 | 3/2/2 | 4/1/2 | 3/1/3 |

Results in HV values are reported in Table 7 and Fig. 7. A larger HV value indicates a better optimization result.

## E.4 PERFORMANCE ON EXPENSIVE MULTI-OBJECTIVE OPTIMIZATION UNDER THE SAME EVALUATION BUDGET

The statistical test results reported in the last row of Table 5 show that ParEGO Knowles (2006) and OREA Yu et al. (2019) are the best two comparison algorithms when compared with our MOEA/D-FS. In this subsection, we evaluate the performance of MOEA/D-FS when no extra evaluation is saved. For this purpose, we compare the optimization performance of these three SAEAs under the same evaluation budget: 10 evaluations ($1d$) for surrogate initialization and 50 evaluations for further optimization. The statistical test results are reported in Table 8. It can be seen that our MOEA/D-FS generally outperforms the compared SAEAs when only $1d$ evaluations are used to

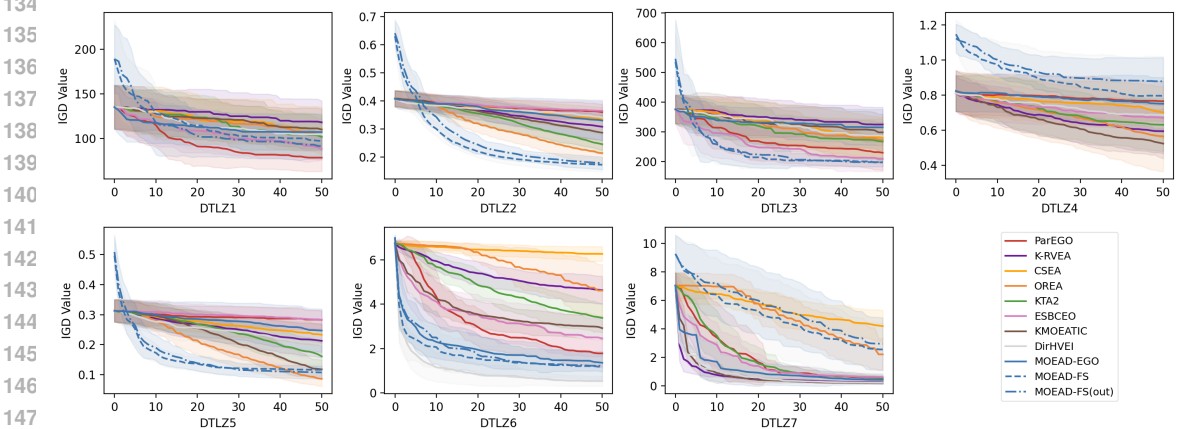

Figure 6: IGD curves averaged over 30 runs on 7 DTLZ problems. Solid lines are mean values, while shadows are error regions. MOEA/D-FSs and comparison algorithms initialize their surrogates with 10, 100 samples, respectively. Extra 50 evaluations are allowed in the further optimization. Note that 'FS(out)' indicates the target task is excluded from the range of related tasks during the meta-learning procedure). X-axis denotes the number of evaluations used after the surrogate initialization.

Table 7: Mean HV values and standard deviation (in parentheses) obtained from 30 runs on 7 DTLZ problems. MOEA/D-FS and comparison algorithms initialize their surrogates with 10, 100 samples, respectively. Extra 50 evaluations are allowed in the further optimization. '+', '≈', and '−' denote MOEA/D-FS is statistically significantly superior to, almost equivalent to, and inferior to the compared algorithms in the Wilcoxon rank sum test (significance level is 0.05), respectively. The last row counts the total win/tie/loss results.

| Problems | MOEAD-EGO | MOEAD-FS | ParEGO | K-RVEA | KTA2 | CSEA | OREA | ESBCEO | KMOEATIC | DirHVEI |
|---|---|---|---|---|---|---|---|---|---|---|
| DTLZ1 | 0.00e+0(0.00e+0)≈ | 0.00e+0(0.00e+0) | 0.00e+0(0.00e+0)≈ | 0.00e+0(0.00e+0)≈ | 0.00e+0(0.00e+0)≈ | 0.00e+0(0.00e+0)≈ | 0.00e+0(0.00e+0)≈ | 0.00e+0(0.00e+0)≈ | 0.00e+0(0.00e+0)≈ | 0.00e+0(0.00e+0)≈ |
| DTLZ2 | 2.02e-1(1.28e-1)+ | 4.69e-1(3.70e-2) | 1.21e-1(4.31e-2)+ | 1.93e-1(8.93e-2)+ | 3.19e-1(6.49e-2)+ | 1.59e-1(5.39e-2)+ | 3.80e-1(7.64e-2)+ | 1.39e-1(4.55e-2)+ | 2.91e-1(1.29e-1)+ | 2.25e-1(1.04e-1)+ |
| DTLZ3 | 0.00e+0(0.00e+0)≈ | 0.00e+0(0.00e+0) | 0.00e+0(0.00e+0)≈ | 0.00e+0(0.00e+0)≈ | 0.00e+0(0.00e+0)≈ | 0.00e+0(0.00e+0)≈ | 0.00e+0(0.00e+0)≈ | 0.00e+0(0.00e+0)≈ | 0.00e+0(0.00e+0)≈ | 0.00e+0(0.00e+0)≈ |
| DTLZ4 | 6.25e-2(5.53e-2)+ | 1.43e-1(7.17e-2) | 2.03e-2(2.71e-2)+ | 3.81e-2(4.24e-2)+ | 6.49e-2(7.42e-2)+ | 4.30e-2(5.29e-2)+ | 2.11e-1(1.37e-1)≈ | 2.27e-2(2.65e-2)+ | 6.50e-2(7.07e-2)+ | 5.66e-2(5.56e-2)+ |
| DTLZ5 | 4.50e-2(4.17e-2)+ | 1.63e-1(1.60e-2) | 1.29e-2(1.30e-2)+ | 4.82e-2(2.78e-2)+ | 7.98e-2(3.80e-2)+ | 3.08e-2(1.61e-2)+ | 1.49e-2(2.88e-2)+ | 1.64e-2(1.42e-2)+ | 1.58e-1(3.69e-2)+ | 1.08e-1(4.65e-2)+ |
| DTLZ6 | 1.24e-3(3.77e-3)+ | 1.59e-2(3.46e-2) | 2.02e-5(1.09e-4)+ | 0.00e+0(0.00e+0)+ | 0.00e+0(0.00e+0)+ | 0.00e+0(0.00e+0)+ | 0.00e+0(0.00e+0)+ | 0.00e+0(0.00e+0)+ | 0.00e+0(0.00e+0)+ | 9.00e-2(5.84e-2)− |
| DTLZ7 | 3.83e-1(8.50e-2)− | 1.12e-1(1.27e-1) | 2.21e-1(9.60e-2)− | 3.79e-1(2.61e-2)− | 3.70e-1(3.88e-2)− | 3.14e-5(1.69e-4)+ | 2.97e-2(4.45e-2)≈ | 1.71e-1(8.33e-2)− | 4.67e-1(1.27e-2)− | 5.11e-1(8.24e-3)− |
| +/≈/− | 3/3/1 | -/-/- | 3/3/1 | 3/3/1 | 3/3/1 | 4/3/0 | 1/6/0 | 3/3/1 | 2/4/1 | 3/2/2 |

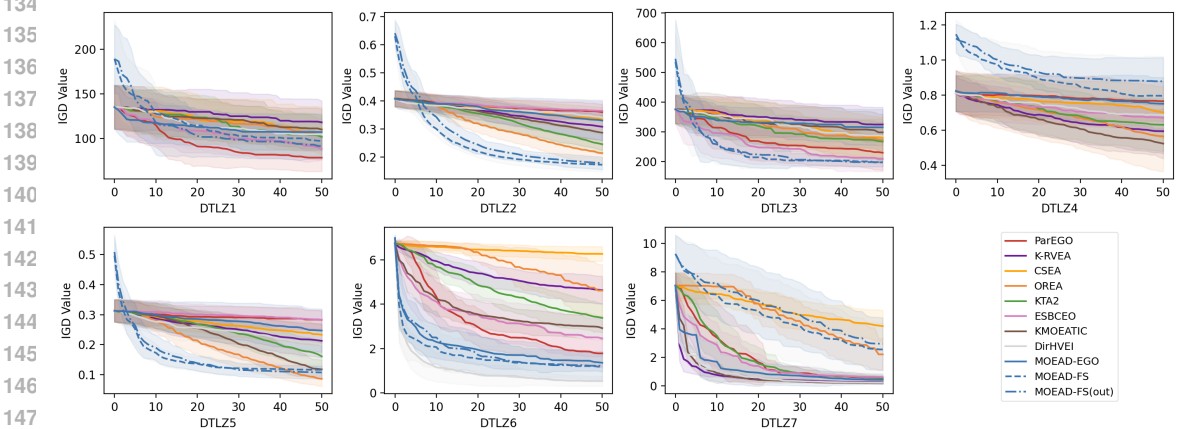

Figure 7: HV curves averaged over 30 runs on 7 DTLZ problems. Solid lines are mean values, while shadows are error regions. MOEA/D-FSs and comparison algorithms initialize their surrogates with 10, 100 samples, respectively. Extra 50 evaluations are allowed in the further optimization. Note that 'FS(out)' indicates the target task is excluded from the range of related tasks during the meta-learning procedure). X-axis denotes the number of evaluations used after the surrogate initialization.

Table 8: Mean IGD+ values and standard deviation (in parentheses) obtained from 30 runs on DTLZ problems. MOEA/D-FS is compared with ParEGO and OREA under the same evaluation budget: 10 evaluations for surrogate initialization and 50 evaluations for the optimization process. '+', '≈', and '−' denote MOEA/D-FS is statistically significantly superior to, almost equivalent to, and inferior to the compared two algorithms in the Wilcoxon rank sum test (significance level is 0.05), respectively. The last row counts the total win/tie/loss results.

| Problem | MOEA/D-FS | ParEGO | OREA |
|---|---|---|---|
| DTLZ1 | 9.70e+1(1.87e+1) | 6.70e+1(4.75e+0)− | 1.10e+2(3.65e+1)≈ |
| DTLZ2 | 1.43e-1(2.29e-2) | 5.51e-1(5.37e-2)+ | 4.28e-1(6.68e-2)+ |
| DTLZ3 | 1.97e+2 (1.64e+1) | 1.84e+2(8.86e+0)≈ | 2.72e+2(6.59e+1)+ |
| DTLZ4 | 4.44e-1(1.35e-1) | 6.29e-1(7.99e-2)+ | 6.45e-1(1.24e-1)+ |
| DTLZ5 | 1.13e-1(2.24e-2) | 4.32e-1(8.88e-2)+ | 3.02e-1(7.63e-2)+ |
| DTLZ6 | 1.11e+0(5.71e-1) | 1.03e+0(4.78e-1)≈ | 5.71e+0(6.73e-1)+ |
| DTLZ7 | 2.47e+0(1.89e+0) | 4.38e-1(1.39e-1)− | 7.12e+0(1.77e+0)+ |
| $+/\approx/-$ | -/-/- | 3/2/2 | 6/1/0 |

Table 9: Mean IGD+ values and standard deviation (in parentheses) obtained from 30 runs on DTLZ problems. Both MOEA/D-FSs initialize their surrogates with 10 samples, extra 50 evaluations are allowed in the further optimization. The last two rows count the statistical test results between MOEA/D-FSs and other compared algorithms.

| MOEA/D-FSs | In-range | Out-of-range |
|---|---|---|
| DTLZ1 | 9.70e+1(1.87e+1)≈ | 9.11e+1(1.53e+1) |
| DTLZ2 | 1.43e-1(2.29e-2)≈ | 1.41e-1(1.75e-2) |
| DTLZ3 | 1.97e+2 (1.64e+1)≈ | 1.98e+1(1.51e+1) |
| DTLZ4 | 4.44e-1(1.35e-1)≈ | 4.96e-1(8.63e-2) |
| DTLZ5 | 1.13e-1(2.24e-2)≈ | 1.03e-1(2.39e-2) |
| DTLZ6 | 1.11e+0(5.71e-1)≈ | 1.17e+0(6.88e-1) |
| DTLZ7 | 2.47e+0(1.89e+0)≈ | 2.86e+0(1.87e+0) |
| $+/\approx/-$ | 0/7/0 | -/-/- |
| vs MOEA/D-EGO | 4/2/1 | 4/2/1 |
| vs 6 Comparisons | 26/9/7 | 27/7/8 |

initialize their surrogates. The effectiveness of our FSEO framework has been demonstrated. Note that OREA is an evolutionary algorithm assisted by ordinal-regression-based surrogates. Currently, our FSEO framework is applicable to the SAEAs working with regression-based surrogates. The meta-learning of ordinal-regression models can be a topic of further research.

# F  ABLATION STUDY: INFLUENCE OF TASK SIMILARITY

In real-world applications, it is optimistic to assume that some related tasks are very similar to the target task. A more common situation is that all related tasks have limited similarity to the target task. To investigate the relationship between task similarity and FSEO optimization performance, we also test the performance in an 'out-of-range' situation, where the original DTLZ is excluded from the range of DTLZ variants during the MDKL meta-learning procedure. As a result, only the DTLZ variants that are quite different from the original DTLZ problem can be used to learn experience. The 'out-of-range' situation eliminates the probability that MDKL surrogates benefit greatly from the DTLZ variants that are very similar to the original DTLZ problem. Detailed definitions of the related tasks used in the 'out-of-range' situation are given in Appendix C. Apart from the related tasks used, the remaining experimental setups are the same as the setups described in Section 5.1. For the sake of convenience, we denote the situation we tested in Section 5.1 as 'in-range' below.

The statistical test results reported in Table 9 show that the 'out-of-range' situation achieves competitive IGD+ values to the 'in-range' situation on all 7 test instances. This suggests that related tasks that are very similar to the target task have a limited impact on the optimization performance of our FSEO framework. Useful experience can be learned from related tasks that are not very similar to the target task. Crucially, when comparing the performance of the 'out-of-range' situation and that of MOEA/D-EGO, we can still observe competitive or improved optimization results on 6

Table 10: Mean IGD+ values and standard deviation (in parentheses) obtained from 30 runs on DTLZ problems. 10 samples are used for initialization and extra 50 evaluations are allowed in the further optimization. $|D_m|$ is the size of the dataset collected from each related task.

| Problem | In-range | | Out-of-range | |
|---|---|---|---|---|
| | $|D_m|$=20 | $|D_m|$=60 | $|D_m|$=20 | $|D_m|$=60 |
| DTLZ1 | 9.70e+1(1.87e+1)$\approx$ | 9.77e+1(1.73e+1) | 9.11e+1(1.53e+1)$\approx$ | 9.93e+1(1.87e+1) |
| DTLZ2 | 1.43e-1(2.29e-2)+ | 1.24e-1(2.11e-2) | 1.41e-1(1.75e-2)+ | 1.29e-1(2.36e-2) |
| DTLZ3 | 1.97e+2 (1.64e+1)$\approx$ | 1.98e+2 (2.21e+1) | 1.98e+1(1.51e+1)$\approx$ | 1.93e+2(1.19e+1) |
| DTLZ4 | 4.44e-1(1.35e-1)$\approx$ | 5.17e-1(5.68e-2) | 4.96e-1(8.63e-2)$\approx$ | 5.17e-1(5.38e-2) |
| DTLZ5 | 1.13e-1(2.24e-2)+ | 9.96e-2(2.18e-2) | 1.03e-1(2.39e-2)$\approx$ | 1.05e-1(2.73e-2) |
| DTLZ6 | 1.11e+0(5.71e-1)$\approx$ | 1.04e+0(6.06e-1) | 1.17e+0(6.88e-1)$\approx$ | 1.22e+0(6.41e-1) |
| DTLZ7 | 2.47e+0(1.89e+0)+ | 7.49e-1(2.61e-1) | 2.86e+0(1.87e+0)+ | 6.96e-1(2.41e-1) |
| $+/\approx/-$ | 3/4/0 | -/-/- | 2/5/0 | -/-/- |

DTLZ problems (see Table 9, the row titled by 'vs MOEA/D-EGO', or Fig. 3). Moreover, it can be seen from the last row of Table 9 that the 'out-of-range' situation achieves better/competitive/worse IGD+ values than all compared SAEAs on 27/7/8 test instances. In comparison, the corresponding statistical test results for the 'in-range' situation are 26/9/7. The difference between these statistical test results is not significant.

A study on the 'out-of-range' situation in the context of extremely expensive multi-objective optimization is presented in Appendix H.2. Consistent results can be observed from Table 12 and Fig. 8.

Consequently, related tasks that are very similar to the target task are not essential to the optimization performance of our FSEO framework. In the 'out-of-range' situation, our MOEA/D-FS can still achieve competitive or better optimization results than MOEA/D-EGO while using only $1d$ samples for surrogate initialization.

# G   ABLATION STUDY: INFLUENCE OF THE SIZE OF DATASETS USED IN META-LEARNING

We also investigated the performance of our FSEO framework when different sizes of datasets $|D_m|$ are used in the meta-learning procedure. The experimental setups are the same as the setups of MOEA/D-FS in Section 5.1 except for $|D_m|$.

It is evident from Table 10 that when each DTLZ variant provides $|D_m| = 60$ samples for the meta-learning of MDKL surrogates, the performance of both MOEA/D-FSs are improved on 2 or 3 DTLZ problems. Particularly, a significant improvement can be observed from the optimization results of DTLZ7. As we discussed in Section 5.1, the poor performance of our experience-based optimization on DTLZ7 is caused by the small size of $D_m$. Optimal solutions have few chances to be included in a small $D_m$, which makes $D_m$ fails to provide the experience about the discontinuity of optimal regions. In comparison, the experience of 'optimal regions' can be learned from large datasets $D_m$ and thus the optimization results are improved significantly.

In conclusion, for our FSEO framework, a large $D_m$ for the meta-learning procedure indicates more useful experience can be learned from related tasks, which further improves the performance of experience-based optimization. Therefore, when applying our FSEO framework to real-world optimization problems, it is preferable to collect more data from related tasks for experience learning.

# H   EXPERIMENTS ON EXTREMELY EXPENSIVE MULTI-OBJECTIVE OPTIMIZATION

In this section, we investigate the performance of our FSEO framework in the context of extremely expensive optimization, where the allowed fitness evaluations on target problems are fewer than that in the experiment carried out in Sections 5.1 of the main file and Appendix F.

Table 11: Mean IGD+ values and standard deviation (in parentheses) obtained from 30 runs on the DTLZ problems. MOEA/D-FS and the comparison algorithms initialize their surrogates with 10, 60 samples, respectively. Extra 30 evaluations are allowed in the further optimization. '+', '≈', and '−' denote MOEA/D-FS is statistically significantly superior to, equivalent to, and inferior to the compared algorithms in the Wilcoxon rank sum test (significance level is 0.05), respectively. The last row is the total win/tie/loss results. Performance improvement can be observed from the comparisons between MOEA/D-FS and MOEA/D-EGO, while 50 evaluations are saved from surrogate initialization.

| Problems | MOEAD-EGO | MOEAD-FS (ours) | ParEGO | K-RVEA | KTA2 | CSEA | OREA | ESBCEO | KMOEATIC | DirHVEI |
|---|---|---|---|---|---|---|---|---|---|---|
| DTLZ1 | 1.07e+2(2.73e+1)≈ | 1.03e+2(2.34e+1) | 8.70e+1(2.53e+1)− | 1.22e+2(3.20e+1)+ | 1.15e+2(3.03e+1)≈ | 1.08e+2(2.64e+1)≈ | 1.11e+2(2.25e+1)+ | 1.00e+2(2.07e+1)≈ | 1.20e+2(2.71e+1)+ | 9.80e+1(2.10e+1)≈ |
| DTLZ2 | 3.49e-1(5.82e-2)+ | 1.57e-1(2.29e-2) | 3.51e-1(5.01e-2)+ | 3.72e-1(4.32e-2)+ | 3.57e-1(4.60e-2)+ | 3.55e-1(5.14e-2)+ | 3.14e-1(3.76e-2)+ | 3.83e-1(3.83e-2)+ | 3.79e-1(4.46e-2)+ | 3.81e-1(4.25e-2)+ |
| DTLZ3 | 3.07e+2(5.32e+1)+ | 2.03e+2(2.42e+1) | 2.16e+2(4.89e+1)≈ | 3.53e+2(7.76e+1)+ | 3.23e+2(8.67e+1)+ | 3.35e+2(6.83e+1)+ | 3.39e+2(7.72e+1)+ | 2.41e+2(5.51e+1)+ | 3.27e+2(8.10e+1)+ | 2.75e+2(6.57e+1)+ |
| DTLZ4 | 5.45e-1(1.09e-1)≈ | 4.91e-1(1.24e-1) | 6.36e-1(8.67e-2)+ | 5.53e-1(9.79e-2)≈ | 5.47e-1(1.02e-1)≈ | 5.84e-1(9.59e-2)+ | 5.14e-1(1.21e-1)≈ | 5.47e-1(7.55e-2)≈ | 4.53e-1(1.03e-1)≈ | 5.23e-1(1.36e-1)≈ |
| DTLZ5 | 2.79e-1(5.69e-2)+ | 1.18e-1(1.25e-2) | 2.78e-1(5.59e-2)+ | 2.82e-1(5.42e-2)+ | 2.60e-1(5.50e-2)+ | 2.77e-1(4.34e-2)+ | 1.99e-1(4.53e-2)+ | 2.94e-1(4.92e-2)+ | 2.69e-1(6.11e-2)+ | 2.75e-1(6.28e-2)+ |
| DTLZ6 | 2.04e+0(7.33e-1)+ | 1.29e+0(6.44e-1) | 2.47e+0(7.39e-1)+ | 5.23e+0(6.17e-1)+ | 4.58e+0(6.36e-1)+ | 6.44e+0(3.53e-1)+ | 5.79e+0(6.70e-1)+ | 3.04e+0(9.46e-1)+ | 3.55e+0(6.90e-1)+ | 6.35e-1(4.37e-1)− |
| DTLZ7 | 1.90e+0(9.19e-1)− | 4.16e+0(2.54e+0) | 1.39e+0(1.49e+0)− | 3.13e-1(6.07e-2)− | 2.05e+0(2.16e+0)− | 5.47e+0(1.31e+0)+ | 5.51e+0(1.32e+0)+ | 9.57e-1(5.40e-1)− | 2.68e-1(1.47e-1)− | 2.79e-1(1.17e-1)− |
| +/≈/− | 4/2/1 | -/-/- | 4/1/2 | 5/1/1 | 4/2/1 | 6/1/0 | 6/1/0 | 4/2/1 | 5/1/1 | 3/2/2 |

## H.1 PERFORMANCE BETWEEN COMPARISON ALGORITHMS

We conduct the experiment described in Section 5.1 of the main file, but with a smaller evaluation budget than the budget listed in Table 2. The size of the initial dataset $S_*$ is set to 10, 60 for our MOEA/D-FS and comparison algorithms, respectively. 30 extra evaluations for further optimization are allowed. The total evaluation budget is 40, 90 for our MOEA/D-FS and comparison algorithms, respectively.

The aim of this subsection is to answer the question below:

- Is our FSEO framework more suitable for the optimization problems in which evaluations are extremely expensive? In other words, will the advantage of our FSEO framework become more prominent if the optimization problems allow a smaller evaluation budget?

The comparison results reported in Fig. 8 and Table 11 show that MOEA/D-FS has achieved competitive or smaller IGD+ values than MOEA/D-EGO on all DTLZ problems except for DTLZ7. Meanwhile, $5d$ evaluations have been saved.

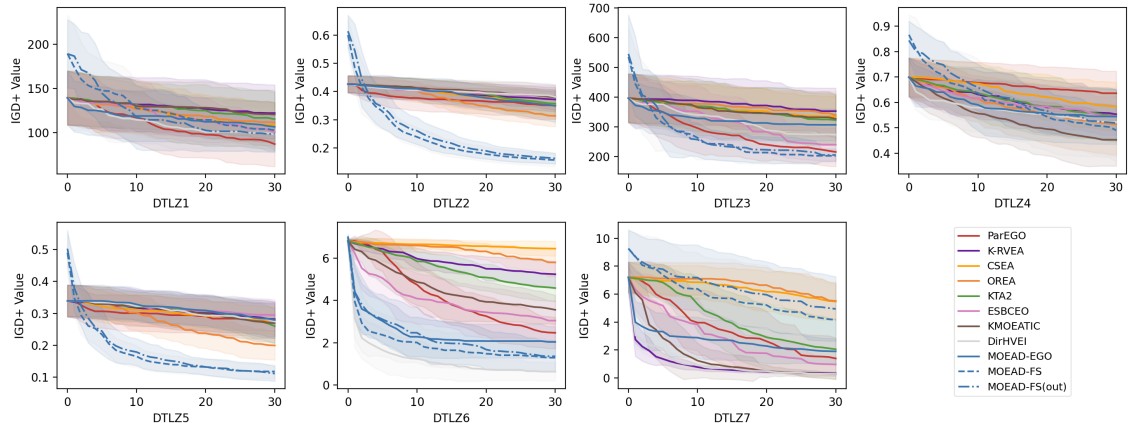

Figure 8: IGD+ curves averaged over 30 runs on 7 DTLZ problems. Solid lines are mean values, while shadows are error regions. MOEA/D-FSs and comparison algorithms initialize their surrogates with 10, 60 samples, respectively. Extra 30 evaluations are allowed in the further optimization. Note that 'FS(out)' indicates the target task is excluded from the range of related tasks during the meta-learning procedure. X-axis denotes the number of evaluations used after the surrogate initialization. In comparison to MOEA/D-EGO, both MOEA/D-FSs achieve smaller or competitive IGD+ values on all DTLZ test problems except for DTLZ7, while 50 evaluations are saved with the assistance from related tasks. Moreover, MOEA/D-FSs achieve the smallest IGD+ values on DTLZ2, DTLZ3, and DTLZ5.

Table 12: Mean IGD+ values and standard deviation (in parentheses) obtained from 30 runs on DTLZ problems. 'Out-of-range' indicates the target task is excluded from the range of related tasks during the meta-learning procedure. Both MOEA/D-FSs initialize their surrogates with 10 samples, extra 30 evaluations are allowed in the further optimization. '+', '≈', and '−' denote the result of the 'out-of-range' situation is statistically significantly superior to, almost equivalent to, and inferior to that of the 'in-range' situation in the Wilcoxon rank sum test (significance level is 0.05), respectively. The last two rows count the statistical test results between MOEA/D-FSs and other compared algorithms.

| MOEA/D-FSs | In-range | Out-of-range |
|---|---|---|
| DTLZ1 | 1.03e+2(2.34e+1)≈ | 9.84e+1(2.04e+1) |
| DTLZ2 | 1.57e-1(2.29e-2)≈ | 1.62e-1(1.90e-2) |
| DTLZ3 | 2.03e+2(2.42e+1)≈ | 2.06e+2(2.13e+1) |
| DTLZ4 | 4.91e-1(1.24e-1)≈ | 5.20e-1(6.92e-2) |
| DTLZ5 | 1.18e-1(2.25e-2)+ | 1.11e-1(2.41e-2) |
| DTLZ6 | 1.29e+0(6.44e-1)≈ | 1.36e+0(7.36e-1) |
| DTLZ7 | 4.16e+0(2.54e+0)≈ | 4.94e+0(2.31e+0) |
| $+/\approx/-$ | 0/7/0 | -/-/- |
| vs MOEA/D-EGO | 4/2/1 | 4/2/1 |
| vs 6 Comparisons | 29/8/5 | 28/9/5 |

Consistent with the results discussed in Section 5.1 of the main file, MOEA/D-FS fails to achieve a competitive result compared to MOEA/D-EGO on DTLZ7 since experience is learned from small datasets collected from related tasks. Although we set a different evaluation budget for all SAEAs, the size of datasets for meta-learning $|D_m|$ has not been modified. However, it can be observed from the statistical test results (see the last row of Tables 5 and 11) that our MOEA/D-FS outperforms the comparison algorithms on 36, 41 test instances when the total evaluation budget of comparison algorithms is set to 150, 90, respectively. This answers the question we raised before: The advantage of our FSEO framework is more prominent in the extremely expensive problems where a smaller evaluation budget is allowed. The comparison between the results obtained from Tables 5 and 11 has demonstrated that our FSEO framework is preferable when solving optimization problems within a very limited evaluation budget.

## H.2 OUT-OF-RANGE ANALYSIS ON EXTREMELY EXPENSIVE OPTIMIZATION

In Section F of the main file, we carried out an experiment to study the influence of task similarity on the performance of experience-based expensive multi-objective optimization. The optimization results obtained from the 'in-range' and the 'out-of-range' situations are compared. In this subsection, we conduct an experiment to investigate the difference between the 'in-range' and the 'out-of-range' situations for extremely expensive multi-objective optimization. The experimental setups are the same as the setups described in Section F of the main file, except the allowed fitness evaluation budget is the same as described in Appendix H.1.

Table 12 gives the statistical test results, it can be seen that the 'out-of-range' situation achieves competitive IGD+ values to the 'in-range' situation on all 7 test instances. In comparison to MOEA/D-EGO, the experience gained in the 'out-of-range' situation leads to competitive or smaller IGD+ values on 6 DTLZ problems. Furthermore, similar results can be observed in the last row of Table 12, the 'out-of-range' situation achieves better/competitive/worse IGD+ values than all compared SAEAs on 28/9/5 test instances. In comparison, the 'in-range' situation achieves better/competitive/worse IGD+ values than all compared SAEAs on 29/8/5 test instances. There is only a minor difference between the optimization results obtained in two situations. These observations are consistent with the conclusions we made in Section F of the main file.

## H.3 RESULT TABLES AND FIGURES IN IGD AND HV METRICS

Results in IGD values are reported in Table 13 and Fig. 9. A smaller IGD value indicates a better optimization result.

Table 13: Mean IGD values and standard deviation (in parentheses) obtained from 30 runs on 7 DTLZ problems. MOEA/D-FS and comparison algorithms initialize their surrogates with 10, 60 samples, respectively. Extra 30 evaluations are allowed in the further optimization. '+', '≈', and '−' denote MOEA/D-FS is statistically significantly superior to, almost equivalent to, and inferior to the compared algorithms in the Wilcoxon rank sum test (significance level is 0.05), respectively. The last row counts the total win/tie/loss results.

| Problems | MOEAD-EGO | MOEAD-FS | ParEGO | K-RVEA | KTA2 | CSEA | OREA | ESBCEO | KMOEATIC | DirHVEI |
|---|---|---|---|---|---|---|---|---|---|---|
| DTLZ1 | 1.07e+2(2.73e+1)≈ | 1.03e+2(2.34e+1) | 8.70e+1(2.53e+1)− | 1.22e+2(3.20e+1)+ | 1.15e+2(3.03e+1)≈ | 1.08e+2(2.64e+1)≈ | 1.11e+2(2.25e+1)+ | 1.00e+2(2.07e+1)≈ | 1.20e+2(2.71e+1)+ | 9.80e+1(2.10e+1)≈ |
| DTLZ2 | 3.69e-1(5.72e-2)+ | 1.91e-1(2.19e-2) | 3.95e-1(3.35e-2)+ | 3.96e-1(3.55e-2)+ | 3.80e-1(4.24e-2)+ | 3.84e-1(4.05e-2)+ | 3.38e-1(3.44e-2)+ | 4.05e-1(3.07e-2)+ | 4.07e-1(3.85e-2)+ | 4.19e-1(2.95e-2)+ |
| DTLZ3 | 3.07e+2(5.32e+1)+ | 2.03e+2(2.42e+1) | 2.16e+2(4.89e+1)≈ | 3.53e+2(7.76e+1)+ | 3.23e+2(8.67e+1)+ | 3.35e+2(6.83e+1)+ | 3.39e+2(7.72e+1)+ | 2.41e+2(5.51e+1)+ | 3.27e+2(8.10e+1)+ | 2.75e+2(6.57e+1)+ |
| DTLZ4 | 8.36e-1(1.51e-1)≈ | 8.47e-1(1.87e-1) | 9.14e-1(1.22e-1)≈ | 7.28e-1(1.16e-1)− | 7.93e-1(1.49e-1)≈ | 8.41e-1(1.48e-1)≈ | 7.89e-1(1.67e-1)≈ | 7.68e-1(1.21e-1)− | 5.97e-1(1.23e-1)− | 7.46e-1(1.35e-1)− |
| DTLZ5 | 2.88e-1(5.64e-2)+ | 1.22e-1(2.10e-2) | 3.10e-1(4.36e-2)+ | 2.99e-1(5.02e-2)+ | 2.73e-1(5.06e-2)+ | 2.97e-1(3.77e-2)+ | 2.12e-1(4.27e-2)+ | 3.10e-1(4.29e-2)+ | 2.93e-1(5.32e-2)+ | 3.10e-1(4.79e-2)+ |
| DTLZ6 | 2.08e+0(7.16e-1)+ | 1.36e+0(6.03e-1) | 2.54e+0(7.09e-1)+ | 5.24e+0(6.15e-1)+ | 4.58e+0(6.36e-1)+ | 6.45e+0(3.51e-1)+ | 5.79e+0(6.67e-1)+ | 3.10e+0(8.82e-1)+ | 3.57e+0(6.85e-1)+ | 8.84e-1(3.53e-1)− |
| DTLZ7 | 2.02e+0(8.97e-1)− | 4.22e-1(2.52e+0) | 1.53e+0(1.42e+0)− | 4.03e-1(7.19e-2)− | 2.12e+0(2.13e+0)− | 5.49e+0(1.31e+0)+ | 5.53e+0(1.32e+0)+ | 1.02e+0(5.29e-1)− | 3.59e-1(1.49e-1)− | 4.11e-1(1.26e-1)− |
| +/≈/− | 4/2/1 | -/-/- | 3/2/2 | 5/0/2 | 4/2/1 | 5/2/0 | 6/1/0 | 4/1/2 | 5/0/2 | 3/1/3 |

Figure 9: IGD curves averaged over 30 runs on 7 DTLZ problems. Solid lines are mean values, while shadows are error regions. MOEA/D-FSs and comparison algorithms initialize their surrogates with 10, 60 samples, respectively. Extra 30 evaluations are allowed in the further optimization. Note that 'FS(out)' indicates the target task is excluded from the range of related tasks during the meta-learning procedure. X-axis denotes the number of evaluations used after the surrogate initialization.

Results in HV values are reported in Table 14 and Fig. 10. A larger HV value indicates a better optimization result.

Table 14: Mean HV values and standard deviation (in parentheses) obtained from 30 runs on 7 DTLZ problems. MOEA/D-FS and comparison algorithms initialize their surrogates with 10, 60 samples, respectively. Extra 30 evaluations are allowed in the further optimization. '+', '≈', and '−' denote MOEA/D-FS is statistically significantly superior to, almost equivalent to, and inferior to the compared algorithms in the Wilcoxon rank sum test (significance level is 0.05), respectively. The last row counts the total win/tie/loss results.

| Problems | MOEAD-EGO | MOEAD-FS | ParEGO | K-RVEA | KTA2 | CSEA | OREA | ESBCEO | KMOEATIC | DirHVEI |
|---|---|---|---|---|---|---|---|---|---|---|
| DTLZ1 | 0.00e+0(0.00e+0)≈ | 0.00e+0(0.00e+0) | 0.00e+0(0.00e+0)≈ | 0.00e+0(0.00e+0)≈ | 0.00e+0(0.00e+0)≈ | 0.00e+0(0.00e+0)≈ | 0.00e+0(0.00e+0)≈ | 0.00e+0(0.00e+0)≈ | 0.00e+0(0.00e+0)≈ | 0.00e+0(0.00e+0)≈ |
| DTLZ2 | 1.63e-1(8.93e-2)+ | 4.37e-1(3.48e-2) | 9.85e-2(3.44e-2)+ | 9.85e-1(4.43e-2)+ | 1.25e-1(4.84e-2)+ | 1.17e-1(5.59e-2)+ | 1.73e-1(4.75e-2)+ | 1.25e-1(5.19e-2)+ | 9.43e-2(4.62e-2)+ | 7.68e-2(3.26e-2)+ |
| DTLZ3 | 0.00e+0(0.00e+0)≈ | 0.00e+0(0.00e+0) | 0.00e+0(0.00e+0)≈ | 0.00e+0(0.00e+0)≈ | 0.00e+0(0.00e+0)≈ | 0.00e+0(0.00e+0)≈ | 0.00e+0(0.00e+0)≈ | 0.00e+0(0.00e+0)≈ | 0.00e+0(0.00e+0)≈ | 0.00e+0(0.00e+0)≈ |
| DTLZ4 | 6.44e-2(6.93e-2)≈ | 1.00e-1(6.58e-2) | 8.65e-2(3.17e-2)+ | 2.28e-2(4.11e-2)+ | 2.18e-2(3.52e-2)+ | 1.01e-2(2.38e-2)+ | 5.58e-2(6.13e-2)+ | 1.55e-2(2.64e-2)+ | 4.77e-2(5.93e-2)+ | 5.40e-2(7.50e-2)+ |
| DTLZ5 | 2.62e-2(2.64e-2)+ | 1.60e-1(1.54e-2) | 7.89e-3(1.16e-2)+ | 1.51e-2(1.58e-2)+ | 2.60e-2(1.91e-2)+ | 1.08e-2(1.14e-2)+ | 4.57e-2(2.76e-2)+ | 1.43e-2(1.32e-2)+ | 2.04e-2(2.38e-2)+ | 1.01e-2(2.08e-2)+ |
| DTLZ6 | 3.82e-4(2.06e-3)≈ | 1.07e-2(2.64e-2) | 0.00e+0(0.00e+0)≈ | 0.00e+0(0.00e+0)≈ | 0.00e+0(0.00e+0)≈ | 0.00e+0(0.00e+0)≈ | 0.00e+0(0.00e+0)≈ | 8.07e-3(3.03e-2)≈ | 0.00e+0(0.00e+0)≈ | 3.00e-2(4.66e-2)− |
| DTLZ7 | 6.98e-2(1.00e-1)≈ | 4.14e-2(8.25e-2) | 8.22e-2(8.32e-2)− | 2.65e-1(3.94e-2)− | 1.31e-1(1.20e-1)− | 0.00e+0(0.00e+0)≈ | 0.00e+0(0.00e+0)≈ | 8.06e-2(8.74e-2)− | 3.58e-1(4.00e-2)− | 4.15e-1(3.70e-2)− |
| +/≈/− | 2/5/0 | -/-/- | 3/3/1 | 3/3/1 | 3/3/1 | 3/4/0 | 3/4/0 | 3/3/1 | 3/3/1 | 3/2/2 |

## I PERFORMANCE ON REAL-WORLD EMOPs: NETWORK ARCHITECTURE SEARCH

In this section, we demonstrate the performance of our FSEO framework on a real-world EMOP: network architecture search (NAS). NAS is a popular multi-objective optimization problem in machine learning community. We use a NAS benchmark Lu et al. (2023) and consider the architecture

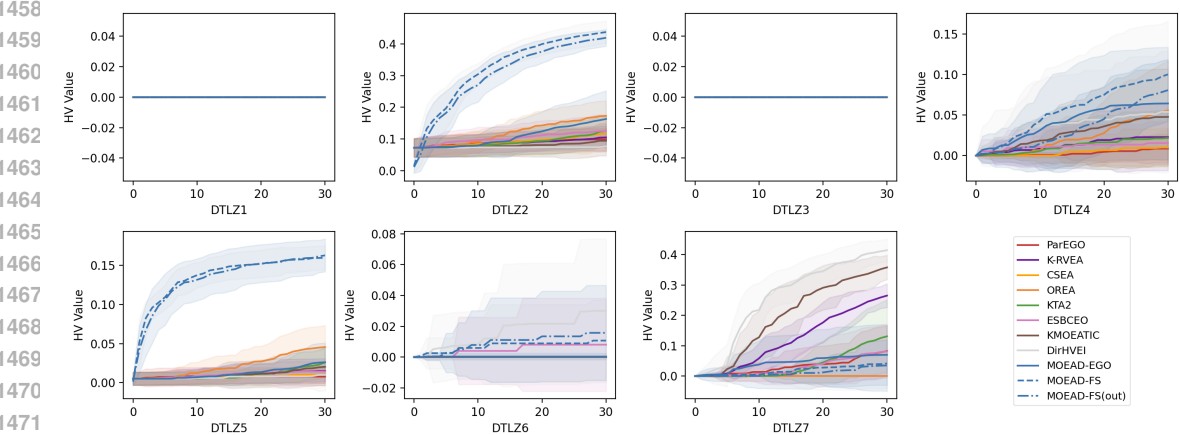

Figure 10: HV curves averaged over 30 runs on 7 DTLZ problems. Solid lines are mean values, while shadows are error regions. MOEA/D-FSs and comparison algorithms initialize their surrogates with 10, 60 samples, respectively. Extra 30 evaluations are allowed in the further optimization. Note that 'FS(out)' indicates the target task is excluded from the range of related tasks during the meta-learning procedure. X-axis denotes the number of evaluations used after the surrogate initialization.

optimization of a Transformer network with two objectives: error and flops. The considered NAS problem has $d$=18 variables, each variable denotes the architecture of a network node. Related tasks are other Transformer architecture optimization problems with different architecture setups. The comparison algorithms are the same as described in Appendix E.1.

## I.1 EXPERIMENTAL SETUPS

In the meta-learning procedure, both the support set and the query set contain 18 data points, thus $|D_m|$=36. For MOEA/D-FS, 18 samples from the target task are used to initialize surrogates, while other comparison algorithms use 100 samples for initialization. For all algorithms, extra 50 evaluations are allowed as optimization budget (the x-axis in Fig. 4). Therefore, the total evaluation budgets for MOEAD-FS and other comparison algorithms are 68 and 150, respectively. The remaining setups are the same as listed in Table 2.

## I.2 OPTIMIZATION RESULTS AND ANALYSIS

Fig. 4 illustrates the optimization results in terms of Hypervolume (HV) values, a large HV value indicates a good performance. We can observe that MOEA/D-FS, K-RVEA, and DirHVEI are preferable to the remaining comparison algorithms in this NAS problem. However, we should note that MOEA/D-FS uses only $1d$ samples from the target task to initialize surrogates. Due to the small initialization dataset, at the early stage, the initial HV value of MOEA/D-FS is smaller than the initial HV values of other comparison algorithms (see Fig 4). With our meta-learning models, MOEA/D-FS adapts to the target task rapidly and it achieves a competitive HV value within 50 additional evaluations. This implies that MOEA/D-FS has saved 82 more evaluations than K-RVEA and DirHVEI by learning experience from related tasks. Therefore, the effectiveness of our FSEO framework on this real-world EMOP is demonstrated.

## J SUMMARY OF EXPERIMENTS

Our computational studies have demonstrated the following: First, we provide empirical evidence to show the effectiveness of learning experience: The meta-learning of neural network parameters and base kernel parameters are essential to the modeling accuracy of a MDKL model. As a result, our MDKL model outperforms the compared meta-learning modeling and non-meta-learning modeling

methods on both the engine fuel consumption regression task and the sinusoid function regression task.

Second, we demonstrate the main contribution of this work via different EMOP benchmarks: In most situations, the proposed FSEO framework can assist regression-based SAEAs to reach competitive or even better optimization results, while the cost of surrogate initialization is only $1d$ samples. Due to the effectiveness of saving evaluations, our FSEO framework is preferable to other SAEAs when solving problems within a very limited evaluation budget. Moreover, some empirical guidelines are concluded to help the application of our FSEO framework. For the influence of task similarity, we find that related tasks that are very similar to the target task are not necessary to the application of our approach. The influence of these similar tasks on the optimization performance is limited. Our FSEO framework can achieve competitive results without datasets from very similar related tasks. Besides, for the related tasks used for meta-learning, we have demonstrated that more useful experience can be learned if more data points are sampled from related tasks.

Third, the effectiveness of our FSEO framework is validated on a real-world engine calibration problem. Competitive or better results are achieved on the objective and constraint functions, while $1d$ samples are used to initialize surrogates. Therefore, our FSEO framework can also be applied to popular optimization scenarios such as constrained optimization, showing the generality and broad applicability of FSEO.

