# OpenReview forum: "FSEO: A Few-Shot Evolutionary Optimization Framework for Expensive Multi-Objective Optimization and Constrained Optimization"
_ICLR.cc/2025/Conference — ICLR 2025 Conference Withdrawn Submission_

### Official Review · Reviewer_4ML5 · 2024-10-27

**Soundness:** 2
**Presentation:** 3
**Contribution:** 2
**Rating:** 5
**Confidence:** 4

**Summary:**

Existing meta-learning-based surrogate models are primarily designed for single-objective expensive optimization problems. Unlike existing approaches, this paper focuses on multi-objective expensive optimization problems (EMOPs) and constrained expensive optimization problems (ECOPs). A novel meta-learning modeling approach is developed to train surrogate models within the few-shot evolutionary optimization (FSEO) framework, along with an accuracy-based update strategy for adjusting the surrogate model during optimization. Experimental results demonstrate the effectiveness of the proposed algorithm.

**Strengths:**

1. This paper innovatively applies meta-learning to expensive multi-objective optimization problems by designing an accuracy-based update strategy to adapt surrogates. This work has some academic value.
2. The experiments in this paper are sufficiently comprehensive, demonstrating the effectiveness of the proposed method from multiple perspectives.

**Weaknesses:**

1. The reasons why the paper considers addressing EMOPs and ECOPs are unclear.
2. A key challenge in ECOPs is finding feasible solutions. This work does not incorporate constraint-handling techniques, and although it finds feasible solutions for a real-world problem with four constraints, the comparison algorithms do so as well, suggesting that the ECOP addressed here is relatively simple. Thus, this does not guarantee that the proposed algorithm will be effective on other ECOPs. Additionally, the title and abstract highlight ECOPs handling as a key focus, which is somewhat misleading.

**Questions:**

1. The paper mentions that meta-learning has already been applied to single-objective expensive optimization tasks. What are the similarities and differences between the proposed method and existing methods?
2. How does the proposed algorithm ensure that the solutions found for ECOPs are feasible?
3. The title mentions that the proposed algorithm can address ECOPs, but only a single example was tested, which is not convincing. Therefore, the title may not be appropriate, or more experiments on ECOPs should be included.
4. This paper is only tested on the DTLZ benchmark suite. The effectiveness of the algorithm should be validated on more benchmark suites.

---

> ### Author Response · Authors · 2024-11-24
>
> Thank you for your thorough review and valuable feedback on our work. The revised paper is attached and revisions are marked in red color.
> --- ---
> $\textbf{W1.}$
> We have revised Sections 1 and 2 to clarify our motivation for focusing on EMOPs and ECOPs. The reasons can be summarized as follows:
>   1. $\textbf{Higher Modeling Requirements}$: We developed a novel meta-learning architecture specifically designed for optimization purposes, which enhances modeling performance in few-shot optimization. Compared to expensive single-objective optimization, EMOPs and ECOPs present a greater challenge due to their higher modeling complexity. This complexity arises from the need to approximate multiple surrogate models for multiple objectives and constraints.
>   2. $\textbf{Relevance and Research Gap}$: EMOPs and ECOPs are widely encountered and highly relevant optimization scenarios. Despite their importance, few studies have explored few-shot optimization for these problems. In contrast, other scenarios, such as large-scale or sparse optimization, are less prioritized compared to EMOPs and ECOPs in terms of their applicability and demand
> --- ---
> $\textbf{W2.}$
> We have revised Sections 1 and 2 to highlight that our FSEO is designed as a general framework. Our FSEO framework aims to enhance sampling efficiency for existing EMOPs and ECOPs through meta-learning experience from related tasks, rather than develop a novel constrained optimization algorithm with specific constraint handling techniques.
>
> To enhance clarity, we have revised the caption of Fig. 1 to explain that the constraint handling technique used in FSEO depends on the underlying constrained SAEA optimizer it used. Additionally, we have revised the first paragraph of Section 5 and Section 5.3 to emphasize that FSEO is an optimization framework aimed at improving sampling efficiency for existing optimization algorithms. Furthermore, we have revised Section 5.3.1 to explain that we have meta-learned surrogates for each objective and each constraint separately for ECOPs. The constraint handling technique is based on the underlying constrained SAEAs we are combine with.
>
> Regarding the experimental results presented in Fig. 5, we have demonstrated that FSEO is compatible with constrained SAEAs and successfully enhances the sampling efficiency of the underlying SAEA (con\_EGO) in both the objective and constraint spaces. These results support our claim in the abstract and at the end of Section 5.3.3. There is no guarantee for the optimization performance of FSEO framework on diverse ECOPs because the underlying constrained SAEAs play an important role in the optimization. However, to ensure the optimization performance on a specific ECOP, we can select a constrained SAEA that is effective on this ECOP and then apply our FSEO framework to this constrained SAEA.
>
> We have gone through our abstract to ensure we are focusing on the performance of FSEO framework on ECOPs instead of developing novel constraint handling techniques. We hope our clarifications and revisions in Sections 1, 2, and 5 are sufficient to solve your concern.

---

> > ### Author Response · Authors · 2024-11-24
> >
> > $\textbf{Q1.}$
> > The similarity between our work and related works is that we all use meta-learning techniques to learn experience from related optimization tasks for solving expensive optimization problems.
> >
> > However, the meta-learning method, meta-learning model architectures, applicable algorithms, and optimization problems to be solved are different.
> > We have revised Section 2 to explain the differences between our FSEO and existing studies:
> >   1. We propose novel architecture of meta-learning model for optimization purpose. Many studies use existing meta-learning models as their surrogates. During the optimization process, these surrogates make predictions with newly observed data, which is a kind of data adaptation rather than model parameter adaptation. The parameters in these models are trained and fixed before the optimization process begins, no further parameter adaptations are made during the optimization since these meta-learning models are originally designed for regression or classification tasks rather than optimization tasks.
> > In comparison, we develop a meta-learning model, MDKL, for optimization purpose. MDKL has novel model architecture with explicit task-specific parameters, which allows continual model parameter adaptations and thus improves modeling performance during the optimization.
> >   2. The generality and broad applicability of FSEO. Existing works are mainly customized for specific algorithms or optimization problems. For example, the meta-learning settings for AFs are not applicable to the SAEAs without AFs. However, our FSEO work on the meta-learning of surrogates and it is applicable to various SAEAs, so our work widens the scope of existing FSO research. A detailed discussion between BO and SAEA is presented in Appendix A.2.
> > In addition, most existing FSO studies investigated only global optimization, leaving other optimization scenarios such as EMOP and ECOP still awaiting for investigation. In contrast, as our MDKL is designed for optimization and is capable of continually adaptation, we pay attention on EMOPs and ECOPs which require more effective models than global optimization.
> >   3. In-depth ablation studies are lacking in the literature, making it unclear which factors affect the performance of FSO. Our extensive ablation studies fill this gap and we conclude some empirical rules to improve the performance of FSO.
> > --- ---
> > $\textbf{Q2.}$
> > Our FSEO framework cannot guarantee the solutions found for ECOPs are all feasible, but it meta-learns experience from related tasks to enhance the efficiency of finding feasible solutions.
> > We revised Section 5.3.1 to explain how our FSEO framework works with constrained SAEAs to find feasible solutions for ECOPs:
> >
> > Specifically, FSEO meta-learns MDKL surrogates for each objective and each constraint separately. For a given underlying constrained SAEA $A$, FSEO adapts the underlying optimization method as well as the constraint handling technique in $A$ as an optimizer, forming a few-shot optimization algorithm $A$-FS. Candidate solutions are firstly evaluated on all MDKL constraint surrogates, based on surrogate predictions and the constraint handling techniques in $A$, potential feasible candidate solutions are selected for expensive evaluations. Our experiment in Section 5.3 have demonstrate that $A$-FS has a higher efficiency than $A$ in terms of finding feasible solutions.
> > --- ---
> > $\textbf{Q3.}$
> > As explained in our response to Weakness 2, we propose a FSEO framework rather than a specific constrained optimization algorithm. Our experiments on ECOPs are designed to demonstrate that our FSEO framework can enhance the sampling efficiency of existing constrained SAEAs in both objective and constraint spaces. Therefore, we use a single real-world ECOP to show the generality and applicability of our FSEO.
> >
> > Our paper title reflects the scope of our contributions without overstating them. Removing EMO and ECO from the title to emphasize the generality of our FSEO framework might create the expectation that our FSEO's performance would be evaluated across all potential optimization scenarios, which is impractical within a single paper. To address this, we use the current title to specify our focus. However, we have revised Sections 1 and 2 to further clarify the scope of our work, which may solve the concern in this comment.
> > --- ---
> > $\textbf{Q4.}$
> > We have added new experiments on real-world network architecture search (NAS) benchmark, which is a set of EMOPs. We report these new experimental setups and results in Section 5.2, Appendix I, and Fig. 4.
> >
> > In addition, in Appendix D, we conduct experiments on a synthetic test problem and a real-world problem to test the modeling performance and the contribution of model components. In Section 5.3, we also conduct experiment on a real-world ECOP.

---

> ### Author Response · Authors · 2024-12-02
>
> Dear Reviewer 4ML5:
>
> Thank you for your reviews.
>
> In our rebuttals, we have added new experiments and discussions as required.
> Regarding the paper title, we can change it to ''FSEO: A General Few-Shot Evolutionary Optimization Framework for Expensive Optimization" to emphasize it is a "General" framework and remove "Constrained Optimization" to avoid any potential misunderstanding.
>
> May we ask if our rebuttals have addressed your concerns?
> Are there any specific issues preventing you from accepting our work?
> If you have any additional questions or concerns, please let us know, and we would be happy to address them based on your suggestions.
>
> Thanks!
>
> Best regards,
> The authors.

---

> > ### Author Response · Authors · 2024-12-03
> >
> > Dear Reviewer 4ML5:
> >
> >
> > The discussion phase will be ending in a few hours.
> > Please let us know if we have addressed your concerns and what you think about the new manuscript title. We look forward to your feedback on our rebuttals.
> >
> >
> > Thanks!
> >
> > Best regards,
> >
> > The authors.

---

### Official Review · Reviewer_jGt8 · 2024-11-01

**Soundness:** 3
**Presentation:** 2
**Contribution:** 3
**Rating:** 6
**Confidence:** 3

**Summary:**

The paper presents a novel Few-Shot Evolutionary Optimization (FSEO) framework that integrates meta-learning with surrogate-assisted evolutionary algorithms to enhance optimization efficiency in expensive multi-objective and constrained optimization scenarios. The approach is innovative and well-motivated, particularly in addressing the gap in existing research, which primarily focuses on single-objective optimization scenarios. Here are some minor suggestions:
1、	The explanation of the meta-learning process and its integration with Gaussian Processes could be further elaborated. Specific details on how the network parameters are adapted during optimization would enhance understanding of the efficacy and mechanics of the proposed method.
2、	While the experiments demonstrate improvements in sampling efficiency, the selection of benchmarks and comparison against state-of-the-art methods need to consider the latest related algorithms.
3、	The discussion section briefly mentions the limitations related to the mathematical definition of related tasks and the framework’s applicability only to regression-based SAEAs. Expanding on these points, possibly with suggestions for future research directions, would provide a more balanced view and potential pathways for advancing the framework.

**Strengths:**

The paper presents a novel Few-Shot Evolutionary Optimization (FSEO) framework that integrates meta-learning with surrogate-assisted evolutionary algorithms to enhance optimization efficiency in expensive multi-objective and constrained optimization scenarios. The approach is innovative and well-motivated, particularly in addressing the gap in existing research, which primarily focuses on single-objective optimization scenarios.

**Weaknesses:**

1）The explanation of the meta-learning process and its integration with Gaussian Processes could be further elaborated. Specific details on how the network parameters are adapted during optimization would enhance understanding of the efficacy and mechanics of the proposed method.
2）While the experiments demonstrate improvements in sampling efficiency, the selection of benchmarks and comparison against state-of-the-art methods need to consider the latest related algorithms.
3）The discussion section briefly mentions the limitations related to the mathematical definition of related tasks and the framework’s applicability only to regression-based SAEAs. Expanding on these points, possibly with suggestions for future research directions, would provide a more balanced view and potential pathways for advancing the framework.

**Questions:**

1）The explanation of the meta-learning process and its integration with Gaussian Processes could be further elaborated. Specific details on how the network parameters are adapted during optimization would enhance understanding of the efficacy and mechanics of the proposed method.
2）While the experiments demonstrate improvements in sampling efficiency, the selection of benchmarks and comparison against state-of-the-art methods need to consider the latest related algorithms.
3）The discussion section briefly mentions the limitations related to the mathematical definition of related tasks and the framework’s applicability only to regression-based SAEAs. Expanding on these points, possibly with suggestions for future research directions, would provide a more balanced view and potential pathways for advancing the framework.

---

> ### Author Response · Authors · 2024-11-24
>
> Thank you for your thorough review and valuable feedback on our work. The revised paper is attached and revisions are marked in red color.
> --- ---
> $\textbf{W1}$
> We have revised Section 4.2 to add more details about the estimation of mean $\mu$ and variance $\sigma$ of the prior distribution in GPs:
> \begin{equation}
> 	\hat{\mu} = \frac{\textbf{1}^T \textbf{R}^{-1} \textbf{y}}{\textbf{1}^T \textbf{R}^{-1} \textbf{1}},
> \end{equation}
> \begin{equation}
> 	\hat{\sigma} = \frac{1}{n} (\textbf{y}-\textbf{1}\hat{\mu})^T\textbf{R}^{-1}(\textbf{y}-\textbf{1}\hat{\mu}).
> \end{equation}
>
> An explanation about how to adapt and integrate our deep kernel with GPs is also provided in Section 4.2:
>
> From Fig. 2, it is clear that task-independent parameters $\mathbf{\gamma^e}$ = $\{\textbf{w}, \textbf{b}, \mathbf{\theta}^e, \textbf{p}^e\}$ are trained on meta data $D_i$. During the optimization process, MDKL adapts task-specific increments $\Delta \mathbf{\theta}^*, \Delta \textbf{p}^*$ (Algorithm 8, line 3) and combines them with experience $\mathbf{\theta}^e$, resulting in task-specific parameters $\mathbf{\theta}^*, \textbf{p}^*$. Hence, the deep kernel parameter $\mathbf{\gamma}^*=\{\textbf{w}, \textbf{b}, \mathbf{\theta}^*, \textbf{p}^*\}$ is available. By invoking Eq. 5, the prior distribution of MDKL is estimated for the following surrogate prediction procedure.
> --- ---
> $\textbf{W2}$
> We have added new experiments on real-world network architecture search (NAS) benchmark, which is a set of newly proposed EMOPs. We report these new experimental setups and results in Section 5.2, Appendix I, and Fig. 4.
>
> In total, we have eight synthetic problems and three real-world problems for modeling, multi-objective optimization, and constrained optimization experiments.
>
> For comparison algorithms, we have included new comparison experiments with a recently proposed MOBO algorithm, DirHVEI, which employs hypervolume-guided composition to address multiple objectives. The experimental results are presented in Figures 3, 4, 6, 7, 8, 9, and 10, as well as in Tables 5, 6, 7, 11, 13, and 14.
>
> In addition, comparison algorithms such as ESBCEO, KMOEATIC, SAB-DE in our experiments are all proposed in the recent year.
> --- ---
> $\textbf{W3}$
> We have revised our limitation and future work in Section 6 as follows, a discussion of the second limitation is available in Appendix B:
>
> The limitations of this work can be summarized as the following two points:
> First, we do not have a mathematical definition of related tasks. As a result, the boundary between related and unrelated tasks is not clear, making it difficult to conduct theoretical analysis on task similarity.
> Second, the proposed framework is currently for regression-based SAEAs only. A detailed discussion on this point is available in Appendix B.
>
> Future work could focus on quantifying task similarity by proposing a metric to measure the similarity between tasks. With an appropriate task similarity measure, systematic studies on few-shot optimization and experience-based optimization could be conducted. In addition, few-shot optimization framework for other SAEA categories can also be a future work.

---

> ### Author Response · Authors · 2024-12-02
>
> Dear Reviewer jGt8:
>
> Thank you for your reviews.
>
> In our rebuttals, we have added the follows as required:
> 1. New experiments on real-world benchmark problems;
> 2. New comparisons with state-of-the-art methods;
> 3. Detailed discussion about limitations.
>
> May we ask if our rebuttals have addressed your concerns?
> If you have any additional questions or concerns, please let us know, and we would be happy to address them based on your suggestions.
>
> Thanks!
>
> Best regards,
> The authors

---

> > ### Author Response · Authors · 2024-12-03
> >
> > Dear Reviewer jGt8:
> >
> > The discussion phase will be ending in a few hours. Please let us know if we have addressed your concerns.
> >
> > In our rebuttals, we have added the follows as required, revised paper is attached:
> > 1. New experiments on real-world benchmark problems: NAS optimization problems.
> > 2. New comparisons with state-of-the-art methods: DirHVEI [1] and qLogHVEI [2].
> > 3. Detailed discussion about limitations.
> > ```
> > [1] Hypervolume-guided decomposition for parallel expensive multi-objective optimization. IEEE Transactions on Evolutionary Computation. 2023.
> > [2] Unexpected Improvements to Expected Improvement for Bayesian Optimization. NeurIPS 2023.
> > ```
> > The new results of qLogHVEI are reported as follows:
> > ```
> > 			MOEA/D-FS				qLogEHVI
> > 		Mean	Min	Std		Mean	Min	Std
> > DTLZ2	1.57e-1	9.92e-2	2.29e-2		2.99e-1	2.15e-1	6.35e-2
> > DTLZ3	2.03e+2	1.60e+2	2.42e+1		1.98e+2	1.76e+2	2.06e+1
> > DTLZ4	4.91e-1	1.97e-1	1.24e-1		3.37e-1	1.90e-1	1.17e-1
> > DTLZ5	1.18e-1	5.84e-2	2.25e-2		2.10e-1	1.19e-1	5.60e-2
> > DTLZ7	4.16e+0	5.86e-1	2.54e+0		4.97e-1	3.00e-1	1.86e-1
> > ```
> >
> > We look forward to your feedback on our rebuttals.
> >
> > Thanks!
> >
> > Best regards,
> >
> > The authors.

---

### Official Review · Reviewer_6zK4 · 2024-11-03

**Soundness:** 2
**Presentation:** 2
**Contribution:** 2
**Rating:** 5
**Confidence:** 4

**Summary:**

This work investigates a meta-learning based few-shot evolutionary optimization (FSEO) approach to improve the performance of surrogate-assisted evolutionary algorithms (SAEA) with a special focus on multi-objective and constrained optimization. It proposes a meta deep kernel learning (MDKL) model as the surrogate model, which combines neural network with Gaussian process. Part of the MDKL model parameters are mete-learned across different tasks, while some parameters (part of GP) are fine-tuned for each specific task. Experimental results show the proposed method can achieve good performance on synthetic and real-world optimization problems.

**Strengths:**

+ This paper is well written and easy to follow.

+ The studied meta-learning based approach is important for real-world expensive optimization.

+ The proposed method achieves good performance on some synthetic and real-world optimization problems.

**Weaknesses:**

**1. Difference between BO and SAEA**

This work claims that the existing works on few-shot optimization are mainly meta-learning based Bayesian optimization (BO) approaches, while this paper focuses on the surrogate-assisted evolutionary algorithm (SAEA). However, the difference between BO and SAEA is not clear to the reader. To my understanding, BO is a general framework for model-based optimization, of which SAEA is a subset that uses an evolutionary algorithm as the search method. For example, the covariance matrix adaptation evolution strategy (CMA-ES) is a popular search method for optimizing the acquisition function in BO.

A detailed explanation of the difference between BO and SAEA is needed.

**2. Novelty and Connection to Related Work**

It seems that meta-learning based Bayesian optimization is already a popular research topic, and different methods have already been proposed for multi-objective optimization [1,2]. A detailed discussion and comparison with these related works are needed.

In addition, in the related work section, this work claims "no further adaptations are made to these surrogates during optimization since they are not originally designed for optimization" for some early work on few-shot Bayesian optimization [3]. However, the surrogate model adaption is a reasonable approach for meta-learning based Bayesian optimization. Does "no further adaptation" still apply to the current meta-learning based BO method?

[1] Speeding Up Multi-Objective Hyperparameter Optimization by Task Similarity-Based Meta-Learning for the Tree-Structured Parzen Estimator,  IJCAI 2023.

[2] BOFormer: Learning to Solve Multi-Objective Bayesian Optimization via Non-Markovian RL, AutoRL@ICML 2024.

[3] Few-shot Bayesian optimization with deep kernel surrogates, ICLR 2021.

**3. Proposed Framework**

- It seems that the proposed few-shot optimization framework is a standard combination of meta-learning and model-based optimization. Compared with existing work, what are the novelty and advantages/disadvantages of the proposed framework and the proposed methods for each step? A detailed ablation study for each algorithm step could also be very helpful for readers to truly understand the contribution of this work.

- This work claims it "focuses on its performance on two common expensive optimization scenarios: multi-objective EOPs (EMOPs) and constrained EOPs (ECOPs)". However, no multi-objective or constrained optimization component has been shown and discussed in the proposed framework. In the experiment section, a popular decomposition-based method (MOEA/D) is used to handle the multi-objective optimization problem. However, it seems that MOEA/D can also be used with other meta-learning based approaches for multi-objective optimization. It is unclear why the proposed framework in this work is more suitable for multi-objective optimization.

**4. Expriments**

- For multi-objective optimization, the proposed framework is only tested on one synthetic test benchmark (DTLZ). More experimental results on real-world multi-objective optimization problems are needed.

- For constrained optimization, one real-world case study is provided, but the details of how the proposed framework deals with the constraints and its novelty/advantages over existing work are missing.

- The proposed framework is only compared with other SAEAs, and the comparison with BO methods is missing.

- Comparison with other meta-learning methods is missing.

**Questions:**

See weaknesses.

---

> ### Author Response · Authors · 2024-11-24
>
> Thank you for your thorough review and valuable feedback on our work. The revised paper is attached and revisions are marked in red color.
>   --- ---
> $\textbf{W1. Difference between BO and SAEA}$.
> We have revised our Section 2 and added the following explanation to Appendix A.2:
> BO and SAEA are both model-based optimization methods for solving expensive optimization problems. The difference between BO and SAEA can be summarized as follows:
> 1. Surrogate models type. BO uses probabilistic models, such as GPs, as surrogates. In comparison, SAEAs are flexible and can use any type of approximation model, not limited to probabilistic models.
> 2. Selection criterion. BO designs an acquisition function (AF) as the selection criterion for candidate solutions, explicitly considering the uncertainty in the probabilistic models. However, SAEAs do not necessarily account for model uncertainty. Instead, they focus on diversity and convergence as selection criteria, which can be implemented through separate functions.
> 3. Search algorithm. BO has no limitation on the search algorithm and can use either gradient-based or gradient-free optimization (such as EAs) to search candidate solutions. In contrast, SAEAs use only EAs as their underlying optimization algorithms.
>
> As a result, there is some overlap between BO and SAEAs. A typical example is ParEGO, which employs GPs as its surrogates and designs an expected improvement (EI) function as its AF to consider uncertainty. Additionally, an EA is used as the underlying search algorithm.
>
> Our FSEO framework focuses on meta-learning surrogates instead of AFs, making it compatible with various SAEAs that do not rely on model uncertainty or AFs as selection criteria. In comparison, existing studies mainly work on the meta-learning of AFs, which limits their generality and applicability to SAEAs.
>
> --- ---
> $\textbf{W2.1. Novelty and Connection to Related Work}$.
> We have carefully reviewed the provided references and discussed their differences from our work as follows:
> $\textbf{Meta-learn TPE}$:
> This method meta-learns acquisition functions (AFs) for the Tree-Structured Parzen Estimator (TPE), which is a variant of BO that uses kernel density estimators (KDEs) instead of GPs as surrogates. Specifically, meta-learn TPE focuses on the task kernel within the AF, while KDEs are directly adopted from existing studies. In comparison, our work focuses on the meta-learning of surrogates rather than AFs. We have developed a novel meta-learning architecture to ensure model parameters can be adapted continually during the optimization, which distinguishes our work from meta-learn TPE and other existing FSO algorithms.
>
> In addition, meta-learn TPE is customized for TPE and cannot be applied to other optimization methods. In contrast, our FSEO is general evolutionary optimization framework, the MDKL model and surrogate management strategy in FSEO are applicable to all regression-based SAEAs. With different underlying SAEAs, our FSEO can solve different expensive optimization problems, such as EMOPs and ECOPs we demonstrated in our experiments, showing greater generalizability than meta-learn TPE.
>
> $\textbf{BOFormer}$:
> This method is a reinforcement learning (RL)-based optimization method, it learns from the history of previous actions and observations to enhance its AF for multi-objective Bayesian Optimization (MOBO). Sequence modeling methods, such as Transformers, are employed to learn its AF from histories.
>
> However, it is important to note that BOFormer is not a meta-learning method -- it does not learn experience from other related tasks, which is a key point to distinguish BOFormer from our work and the related optimization algorithms we discussed in Section 2. Our meta-learning process focuses on the samples collected from related tasks and the adaptation process focuses on the samples collected from the target task. In comparison, BOFormer only focuses on the history of the target work.
>
> In addition, our work title is 'evolutionary optimization framework' but BOFormer uses an RL framework, which is not such relevant to our work. Moreover, our work aim to learn efficient surrogate models, while BOFormer is designed to learn effective AFs.
>
> In our humble opinion, the only similarity between BOFormer and our work is that they both address expensive multi-objective problems (EMOPs). However, EMOPs are just one of the optimization scenarios we substantiated for our framework.
>
> We have revised Sections 1 and 2 to emphasize that our work focuses on meta-learning experiences for constructing effective surrogates and designing a general optimization framework capable of addressing various optimization problems. From this perspective, we have only discussed related experience-based optimization algorithms in Appendix A.1 and Section 2. In contrast, studies on non-experience-based multi-objective or constrained optimization are less relevant to our work.

---

> > ### Author Response · Authors · 2024-11-24
> >
> > $\textbf{W2.2 Novelty and Connection to Related Work}$
> > To the best of our knowledge, 'no further adaptation' still applies to current few-shot BO (FSBO) or meta BO (MBO) methods. We have revised Section 2 to add the following explanations and clarify our contributions regarding model parameter adaptations. The complete revision is available in the updated pdf.
> >
> > Existing works typically adopt surrogate models directly from prior studies. For example, [3] utilized DKT models and customized an underlying optimization algorithm for FSO, while [1] employed KDEs directly and designed a meta-learning setting for AFs. In these approaches, the parameters of surrogate models are trained and fixed before the optimization process begins. Further adaptations are limited to incorporating newly observed data into the prediction process, without updating the surrogate parameters themselves. Differently, in our MDKL, continual adaptations are made on the task-specific parameters. By leveraging newly observed data during optimization, our adapted surrogates produce better predictions toward the target optimization problem.
> > [1] Speeding Up Multi-Objective Hyperparameter Optimization by Task Similarity-Based Meta-Learning for the Tree-Structured Parzen Estimator, IJCAI 2023.
> > [3] Few-shot Bayesian optimization with deep kernel surrogates, ICLR 2021.
> > --- ---
> > $\textbf{W3. Proposed Framework}$
> > $\textbf{Novelties}$. We have revised Sections 1 and 2 to highlight the following novelties:
> > 	Our novelties are the development of new meta-learning model and the design of general few-shot evolutionary optimization framework. Specifically, we propose a novel architecture of meta-learning model for optimization purpose, where parameters are explicitly designed as task-independent parameters and task-specific parameters, respectively. Our meta-learning method pre-trains task-independent parameters to learn common features / experience from related tasks before the optimization of the target task. After that, the optimization process begins, and task-specific parameters are fitted with data that is observed from the target task. The model prediction is determined by task-independent parameters, task-specific parameters, previous observations from the target task, and the solution to be predicted.
> >
> > In comparison, existing works do not have such an well-designed model architecture. Their models do not have explicit task-specific parameters, indicating it is difficult for them to adapt model parameters during the optimization process. As a result, their model adaptations are implemented by introducing the data that is newly observed from the target task, without adaptations on model parameters.
> >
> > In addition, we propose a general evolutionary optimization framework with surrogate management strategy to work with existing SAEAs. Unlike existing works that are customized for specific problems or specific BO, our surrogate management strategy is embedded in a general SAEA framework, making our FSEO compatible with diverse SAEAs and optimization scenarios (Due to the space limitation of a single paper, we substantiated only two popular optimization scenarios: EMOP and ECOP in our work, and we have not claim contributions on other optimization scenarios that we have not tested for now). In contrast, existing works mainly focus on single-objective optimization, while studies on EMOPs and ECOPs are relative limited and customized for BO (especially AFs in BO).
> >
> > $\textbf{Advantages}$. The advantages are the high modeling accuracy of MDKL and great generality of FSEO. Our model architecture allows the model parameters being continually adapted during the optimization, and the FSEO are applicable to SAEAs with different techniques to handle multiple objectives or constraints.
> >
> > $\textbf{Disadvantages}$. The limitation of our work is discussed in Section 6 and Appendix B.
> >
> > $\textbf{Ablation Studies}$. In Appendix D, we conduct ablation studies to evaluate the contributions of individual components in our MDKL. Specifically, we design several variants of our MDKL, each MDKL variant consists of different model components. Experiments are performed on two modeling problems to demonstrate the performance of our MDKL and the contribution of each MDKL component. Experimental setups and results are presented in Appendixes D.1 and D.2, respectively. The comparison between MDKL and its variants show that each component contributes to the overall performance of our algorithm. In the updated pdf, we revised the beginning of Section 5 and the title of Appendix D to highlight the aforesaid experiments and results.
> >
> > In addition, more ablation studies are reported in Section 5.2, Appendixes F and G to investigate the influence of meta data on our algorithm, which are beneficial to the application of our work when solving optimization problems.

---

> > > ### Author Response · Authors · 2024-11-24
> > >
> > > $\textbf{W3.2 Proposed Framework}$
> > > To fully understand our claim, we kindly suggest focusing on the earlier part of the statement:
> > > ``Our FSEO framework is a general framework, but we focus on its performance on EMOPs and ECOPs in this paper.''
> > > Since FSEO is designed as a general framework, our primary consideration is the compatibility across diverse optimization scenarios, rather than the development of specific components customized for EMOPs or ECOPs. For various multi-objective or constrained SAEAs, their methods for handling multiple objectives or constraints are encapsulated within the module `SAEA optimizer' in our diagram Fig. 1, showing the compatibility of our framework with diverse SAEAs. We have revised the caption of Fig. 1 to clarify this.
> > >
> > > We would like to clarify a minor mistake that we use MOEA/D-EGO as an example in our EMOP experiments, not MOEA/D.
> > > We agree with the comment that MOEA/D-EGO could also be used with other meta-learning methods. However, our experiments just use MOEA/D-EGO as an example to demonstrate our compatibility with existing SAEAs and our framework is working for EMOPs. There is no conflict between our work and other studies which might use MOEA/D-EGO with other meta-learning methods.
> > >
> > > Due to the well-designed model architecture and meta-learning method, our model performance is improved and thus is more suitable for solving optimization scenarios that require cooperations between multiple surrogates. That is why we only claim our contributions on two optimization scenarios: EMOPs and ECOPs. These two optimization scenarios need multiple surrogates to approximate either objectives or constraints.
> > > --- ---
> > > $\textbf{W4.1. Experiments}$
> > > We have added a new real-world network architecture search experiment and reported experimental setups and results in Section 5.2, Appendix I, and Fig. 4.
> > > --- ---
> > > $\textbf{W4.2. Experiments}$
> > > We have revised Section 5.3.1 to explain that we have meta-learned surrogates for each objective and each constraint separately for ECOPs. The constraint handling technique is based on the underlying constrained SAEAs we are combine with. The novelty and advantages of our method are explained in our response to Weakness 3.1: We designed a meta-learning with novel architecture, where task-independent parameters are trained with meta data, and task-specific parameters are adapted continually with newly observed data. Additionally, our surrogate management strategy shows how to and when to update surrogates, making it applicable to diverse SAEAs.
> > > --- ---
> > > $\textbf{W4.3. Experiments}$
> > > We have included new comparison experiments with a recently proposed MOBO algorithm, DirHVEI, which employs hypervolume-guided composition to address multiple objectives. The experimental results are presented in Figures 3, 4, 6, 7, 8, 9, and 10, as well as in Tables 5, 6, 7, 11, 13, and 14.
> > >
> > > Additionally, as discussed in our response to Weakness 1, there is some overlap between BO and SAEAs. While all our comparison algorithms are categorized as SAEAs, some also belong to the MOBO category. For instance, ParEGO, MOEA/D-EGO, K-RVEA, OREA, and ESBCEO are all MOBO algorithms, with some (e.g., ESBCEO) being recently proposed. Furthermore, certain constrained SAEAs in our comparisons, such as cons\_EGO, also qualify as constrained BO methods.
> > > --- ---
> > > $\textbf{W4.4. Experiments}$
> > > The comparison with other meta-learning methods is presented in Appendix D as part of our experiments to evaluate the modeling performance. We selected some meta-learning methods that are highly relevant to our approach for the comparison.

---

> > > > ### Comment · Reviewer_6zK4 · 2024-11-26
> > > >
> > > > Thank you for the thorough response. Some of my concerns have been properly addressed and I raise my score to 5.
> > > >
> > > > There are some follow-up questions:
> > > >
> > > > 1. Can you provide a detailed discussion on the comparison between meta-learning acquisition function for BO v.s. meta-learning surrogate function for SAEAs? What are the advantages and disadvantages of these two approaches?
> > > >
> > > > 2. Can we use the method proposed in this work to solve the problems considered in [1,2]? If so, an experimental comparison could be very helpful to show its advantages.
> > > >
> > > > [1] Speeding Up Multi-Objective Hyperparameter Optimization by Task Similarity-Based Meta-Learning for the Tree-Structured Parzen Estimator, IJCAI 2023.
> > > >
> > > > [2] BOFormer: Learning to Solve Multi-Objective Bayesian Optimization via Non-Markovian RL, AutoRL@ICML 2024.
> > > >
> > > > 3. Some state-of-the-art MOBO methods are missing in the experiments, such as qEHVI[3] (and its updated version in [4]). Based on my experience, qEHVI (and its updated version) can significantly outperform many baseline methods in the current experiments for DTLZ.
> > > >
> > > > [3] Differentiable Expected Hypervolume Improvement for Parallel Multi-Objective Bayesian Optimization. NeurIPS 2020.
> > > >
> > > > [4] Unexpected Improvements to Expected Improvement for Bayesian Optimization. NeurIPS 2023.

---

> > > > > ### Author Response · Authors · 2024-11-28
> > > > >
> > > > > Thank you for your thorough review and valuable feedback on our work.
> > > > >
> > > > > --- ---
> > > > > $\textbf{Q1}$:
> > > > > This is a high-level question, as there is a wide range of meta-learning methods targeting either acquisition functions (AFs) or surrogate models. The advantages and disadvantages can vary significantly even between two meta-learning AFs. Therefore, we can only discuss their differences from the perspective of applicability.
> > > > >
> > > > > Meta-learning AFs are specific to Bayesian Optimization (BO) and they are highly relevant to the underlying probabilistic models. Many meta-learning AFs work for only GP-based BO, as GP is one of the most popular models in BO [5]. However, due to the diversity of AFs in the literature, it is often possible to find appropriate AFs and customize them for specifc probabilistic models. For example, [1] developed a customized meta-learning AF for kernel density estimators (KDEs). The customized meta-learning AFs can reach good performance on specific optimization tasks.
> > > > >
> > > > > In contrast, meta-learning models are originally proposed for modeling tasks such as regression and classification rather than for BO. One advantage of meta-learning models is their broad applicability as they have been demonstrated to be effective across diverse fields. Consequently, many few-shot optimization or meta BO studies directly employ existing meta-learning models as their surrogates, even though these models were not originally designed for optimization tasks [6]. In addition, applicability of meta-learning models makes it possible to learning experience for SAEA.
> > > > >
> > > > > Based on the aforesaid discussion, we have developed a meta-learning model with parameters that are continually updated during the optimization. This meta-learning model is designed for optimization purposes and our few-shot evolutionary framework is applicable to SAEAs, which make our unique contributions to the community of expensive optimization.
> > > > >
> > > > > [5] Meta-Learning Acquisition Functions for Transfer Learning in Bayesian Optimization. ICLR 2020.
> > > > >
> > > > > [6] Few-shot Bayesian optimization with deep kernel surrogates, ICLR 2021.

---

> > > > > > ### Author Response · Authors · 2024-11-28
> > > > > > **Follow-up**
> > > > > >
> > > > > > $\textbf{Q2}$:
> > > > > > Yes, [2] conducted experiments on Hyperparameter Optimization (HPO) problems, and [1] conducted experiments on joint Network Architecture Search (NAS) and HPO problems. In comparison, we have conducted a new experiment on NAS problem. The reason that NAS and HPO can be combined in [3] is that they share the same properties: Both NAS and HPO optimize components of given algorithms to improve their performance. The solutions in NAS and HPO problems are encoded in the same way. Therefore, our framework can definitely be used to solve HPO problems.
> > > > > >
> > > > > > However, unlike existing studies which focus solely on EMOPs, our work includes many other experiments on modeling performance, ECOP, and ablation studies on few-shot optimization. The EMOP experiments are only a subset of our comprehensive experimental studies. Therefore, we believe conducting additional extensive EMOP experiments on similar real-world problems would not significantly impact the overall quality of our work.
> > > > > >
> > > > > > --- ---
> > > > > > $\textbf{Q3}$:
> > > > > > We would like to clarify some difference between our work and the studies on MOBO.
> > > > > > 1. Our work presents an evolutionary optimization framework, as we stated in Section 1 that we address EOPs from the perspective of SAEAs rather than BO.
> > > > > >
> > > > > > In our EMOP experiments, we have 9 comparison algorithms, including 5 classic SAEAs representing different categories (regression-based, classification-based, ordinal-regression-based, decomposition-based, and aggregation-based) and 4 state-of-the-art SAEAs (DirHVEI, ESBCEO, KMOEATIC, and KTA2). Notably, most of these 9 comparison algorithms also belong to MOBO and 3 of them are published in the same year as the suggested algorithm qLogEHVI [4]. Additionally, several EHVI-based MOBO have been compared in the paper of DirHVEI. Therefore, although the suggest qEHVI and qLogEHVI outperform some of our 5 classic comparison algorithms, we believe it is unnecessary to add additional MOBO in our EMOP experiments.
> > > > > >
> > > > > > 2. Our experiments on EMOPs aim to demonstrate that our framework can save evaluations / improve sampling efficiency for existing SAEAs while maintaining competitive or enhanced optimization performance, as we explained in the list at the beginning of Section 5. Unlike existing studies that focus solely on EMOPs, the goal of our EMOP experiments is not to outperform specific optimization algorithms but to showcase the framework's capability in enhancing the efficiency of existing SAEAs.
> > > > > >
> > > > > > Therefore, we conduct experiments to show that our framework improves the optimization performance of a classic SAEA while 9$d$ evaluations are saved from optimization budget. In addition, the comparison with other SAEAs or MOBO show that the improvement of optimization performance is significant rather than trivial: MOEA/D-EGO is comparable to DirHVEI after applying our FSEO framework.
> > > > > > If our experiments were designed for outperforming state-of-the-art MOBO, we would just use a state-of-the-art SAEA rather than MOEA/D-EGO as our underlying optimization algorithm. However, if we do so, it would be hard to estimate the significance of optimization performance improvement, as all comparison algorithms would be inferior to our algorithm.
> > > > > >
> > > > > > --- ---
> > > > > > Based on our clarification on the differences between our work and MOBO studies, we hope them are helpful for the reviewer to understand the rationale behind our experimental studies and re-evaluate the overall quality of our work.
> > > > > >
> > > > > > Finally, we have completed the configuration of Botorch environment, as suggested in [4]. We would try our best to present the suggested experimental results (in Q2 and Q3) in textual and tabular form in 6 days.
> > > > > >
> > > > > > Thanks for your help in improving our work quality.
> > > > > >
> > > > > > The authors.

---

> > > > > > > ### Author Response · Authors · 2024-12-03
> > > > > > >
> > > > > > > Dear Reviewer 6zK4:
> > > > > > >
> > > > > > > The discussion phase will be ending in a few hours.
> > > > > > > We hope our previous response have addressed your concerns and clarified the difference between our work and the existing works on MOBO.
> > > > > > >
> > > > > > > We have compared MOEA/D-FS with the suggested MOBO: qLogEHVI, there are IGD+ results obtained from 15 runs:
> > > > > > > ```
> > > > > > > 			MOEA/D-FS				qLogEHVI
> > > > > > > 		Mean	Min	Std		Mean	Min	Std
> > > > > > > DTLZ2	1.57e-1	9.92e-2	2.29e-2		2.99e-1	2.15e-1	6.35e-2
> > > > > > > DTLZ3	2.03e+2	1.60e+2	2.42e+1		1.98e+2	1.76e+2	2.06e+1
> > > > > > > DTLZ4	4.91e-1	1.97e-1	1.24e-1		3.37e-1	1.90e-1	1.17e-1
> > > > > > > DTLZ5	1.18e-1	5.84e-2	2.25e-2		2.10e-1	1.19e-1	5.60e-2
> > > > > > > DTLZ7	4.16e+0	5.86e-1	2.54e+0		4.97e-1	3.00e-1	1.86e-1
> > > > > > > ```
> > > > > > >
> > > > > > > We look forward to your feedback on our rebuttals.
> > > > > > >
> > > > > > > Thanks!
> > > > > > >
> > > > > > > Best regards,
> > > > > > >
> > > > > > > The authors.

---

### Official Review · Reviewer_YwBS · 2024-11-03

**Soundness:** 2
**Presentation:** 2
**Contribution:** 2
**Rating:** 3
**Confidence:** 3

**Summary:**

This paper propose a zero-shot evolutionary framework for expensive MOO and constrained optimization.

**Strengths:**

The studied problems such as MOO, MOEA, MOBO are very hot topics.

**Weaknesses:**

1. **Title**: The title feels overly lengthy and general. Consider refining it to be more concise and specific to the key contribution of the paper.

2. **Section 4.1 - Proposed Framework**: The framework in Section 4.1 appears somewhat ad hoc and lacks a rigorous mathematical foundation. It currently seems more heuristic in nature. Adding formal mathematical justification could strengthen this section.

3. **Suitability for Publication Venues**: The proposed method may be better suited for evolutionary computation venues like *IEEE Transactions on Evolutionary Computation (TEVC)* or *Genetic and Evolutionary Computation Conference (GECCO)*, given its approach and focus.

4. **Equation Formatting**: Equations 2 and 3 take up an unnecessary amount of space. Additionally, consider using `\exp` instead of `exp` to improve the visual consistency of the formulation.

5. **Algorithm 2 - Meta Learning**: The meta-learning approach in Algorithm 2 appears somewhat unconvincing in its current form. It could benefit from a clearer rationale and possibly a refinement of the underlying methodology.

6. **Language and Style**: The paper contains several instances of non-academic language. Tools like Grammarly or ChatGPT could help refine the writing style to meet academic standards.

7. **Zero-Shot Optimization Approach**: The zero-shot approach may not be ideal for handling optimization tasks. For example, if the first case optimizes \( y = x \) and the second optimizes \( y = -x \), both within the domain \([-1, 1]\), it’s unclear how learning from the first case would inform or benefit the second. Consider revisiting this approach.

**Questions:**

See weakness.

---

> ### Author Response · Authors · 2024-11-24
>
> Thank you for your thorough review, the revised paper is attached and revisions are marked in red color.
> --- ---
> $\textbf{W1.}$
> Our key contribution is the development of our general few-shot evolutionary optimization framework, and we test and valid the performance of our framework on expensive multi-objective optimization problems and expensive constrained optimization problems. We think our title has reflected our contributions.
> --- ---
> $\textbf{W2.}$
> Section 4.1 describes the overall workflow of our FSEO framework. To make it easy for readers to understand our work, we encapsulate our work into several steps and modules and provide a diagram to illustrate the framework. Detailed descriptions and mathematical foundations about the modules in Section 4.1 are provided in Sections 4.2 and 4.3.
> --- ---
> $\textbf{W3.}$
> Our work is relevant to two key topics: meta-learning and evolutionary optimization, both of which fall within the scope of ICLR. The most important component of our FSEO framework is the meta-learning of related experience, a contribution firmly rooted in the ML domain, making ICLR an appropriate venue for this work.
>
> In contrast, while venues like TEVC and GECCO focus on evolutionary optimization, our novel contributions in meta-learning methods may receive less attention there.
> --- ---
> $\textbf{W4.}$
> We would move equations 2 and 3 to our Appendix in our final version if we found the space in the main paper were not enough. For now, we have space for equations 2 and 3, and these equations are helpful for readers to understand our model structure.
>
> As for the symbol of $exp$ in equations 2 and 3, we have revised our symbols as suggested.
> --- ---
> $\textbf{W5.}$
> We are unclear what part of our Algorithm 2 is unconvincing to the reviewer, although the architecture of our meta-learning model is different, but the gradient-based training method is a classic and effective way to train meta-learning models in the literature. It would be great if the comment could provide more details about this point so that we can improve our representation.
> --- ---
> $\textbf{W6.}$
> We have gone through our manuscript and refined our writing.
> To avoid potential mistakes, it would be appreciated if the comment can provide some examples of non-academic language.
> --- ---
> $\textbf{W7.}$
> Our few-shot optimization is implemented by meta-learning approach. In the literature, meta-learning models have been demonstrated to be effective to enhancing modeling performance via learning from plenty of related tasks:
> [1]. Chelsea Finn, Pieter Abbeel, and Sergey Levine. Model-agnostic meta-learning for fast adaptation of deep networks. In Proceedings of the 34th International Conference on Machine Learning (ICML'17), 2017.
> [2]. Massimiliano Patacchiola, Jack Turner, Elliot J Crowley, Michael O’Boyle, and Amos Storkey. Bayesian meta-learning for the few-shot setting via deep kernels. In Advance in Neural Information Processing Systems 33 (NeurIPS'20), 2020.
> [3]. Gresa Shala, Thomas Elsken, Frank Hutter, and Josif Grabocka. Transfer NAS with meta-learned bayesian surrogates. In Proceedings of the 11th International Conference on Learning Representations (ICLR'23), 2023.
> [4]. Wenlin Chen, Austin Tripp, and Jose' Miguel Herna ́ndez-Lobato. Meta-learning adaptive deep kernel gaussian processes for molecular property prediction. In Proceedings of the 11th International Conference on Learning Representations (ICLR'23), 2023.
>
> Considering modeling performance plays a key role in model-based optimization, some studies have demonstrated that  using meta-learning models in model-based optimization, namely few-shot optimization, is an effective way to solve expensive optimization problems:
> [1] Martin Wistuba and Josif Grabocka. Few-shot Bayesian optimization with deep kernel surrogates. In Proceedings of the 9th International Conference on Learning Representations (ICLR'21), 2021.
> [2] Shuhei Watanabe, Noor Awad, Masaki Onishi, and Frank Hutter. Speeding up multi-objective hyperparameter optimization by task similarity-based meta-learning for the tree-structured parzen estimator. In Proceedings of the 32nd International Joint Conference on Artificial Intelligence (IJCAI'23), 2023.
> As the performance of few-shot optimization has been demonstrated in the literature, we think it is an ideal approach for optimization tasks.
>
> In addition, few-shot optimization mainly use meta-learning to learn experience from plenty of related tasks. However, the example raised by the comment considers only one related case, which is inappropriate and different from the setting of our work. In the same example but with a meta-learning setting, models can learn the common feature of related cases, such as all cases are linear (i.e. y = x, y = -x), which is beneficial to the further optimization of new unseen case.

---

> ### Author Response · Authors · 2024-11-26
>
> Dear Reviewer YwBS,
>
> Thanks for your response.
>
> We would like to know if our rebuttals have addressed your concerns. If not, would you mind providing us with any follow-up questions to help improve the quality of our work?
>
> Thanks.
>
> Have a nice day.
>
> Best regards,
> The authors.

---

### Note · Authors · 2025-01-22

**Comment:**

Some comments to reviews:
1. It is obvious that all comments from Reviewer YwBS were generated by LLMs such as ChatGPT, which is a clear violation of the ICLR code of conduct. However, Reviewer YwBS was not penalized by ICLR and these low-quality comments were used for making final decision. It is deeply regrettable to receive such irresponsible reviews at ICLR.

2. The advantages of the proposed method over existing meta-learning-based approaches for similar problems have been clarified in our related work and rebuttals.

3. As presented in our title, our work is a few-shot evolutionary optimization FRAMEWORK, not a specific optimization algorithm. While our main concerns are the methodology of few-shot optimization and the compatibility of our framework with common expensive optimization problems. Reviewer 4ML5 only concerns if there are any specific techniques proposed to solve constraints, without any comments on our few-shot optimization method or optimization framework.

4. Our work is a few-shot evolutionary optimization framework and our comparison algorithms are all model-based evolutionary optimization and Bayesian optimization methods.
    4.1. It is incorrect to say that the algorithms mentioned by Reviewer 6zK4 (two Bayesian optimization methods) are more relevant baselines to our work.
    4.2. In addition, we have highlighted the aim of our experiments at the beginning of our experimental studies: Our purpose is to improve the performance of existing algorithms instead of developing a new algorithm that outperforms a specific baseline. For this reason, comparing with the algorithms mentioned by Reviewer 6zK4 would not further validate the paper's effectiveness. Especially when we have already compared with nearly 10 baselines.
    4.3. More importantly, we have already compared the suggested algorithms in our rebuttals, and the real-world problem we added in our rebuttals is more up-to-date than the benchmark suggested by Reviewer 6zK4.

**Withdrawal Confirmation:**

I have read and agree with the venue's withdrawal policy on behalf of myself and my co-authors.